# Remote sensing of emperor penguin abundance and breeding success

Alexander Winterl [1] ✉, Sebastian Richter [1], Aymeric Houstin [1,2,3], Téo Barracho[3,4,5], Matthieu Boureau[3], Clément Cornec[3,6], Douglas Couet [3], Robin Cristofari[3,7], Claire Eiselt[3], Ben Fabry [1], Adélie Krellenstein[3], Christoph Mark[1], Astrid Mainka[1], Delphine Ménard[3], Jennifer Morinay[3], Susie Pottier [3], Elodie Schloesing[3], Céline Le Bohec [3,4,8,9] ✉ & Daniel P. Zitterbart [1,2,9] ✉

Emperor penguins (*Aptenodytes forsteri*) are under increasing environmental pressure. Monitoring colony size and population trends of this Antarctic sea-bird relies primarily on satellite imagery recorded near the end of the breeding season, when light conditions levels are sufficient to capture images, but colony occupancy is highly variable. To correct population estimates for this variability, we develop a phenological model that can predict the number of breeding pairs and fledging chicks, as well as key phenological events such as arrival, hatching and foraging times, from as few as six data points from a single season. The ability to extrapolate occupancy from sparse data makes the model particularly useful for monitoring remotely sensed animal colonies where ground-based population estimates are rare or unavailable.

Emperor penguins (*Aptenodytes forsteri*) breed on land-fast sea ice, making them particularly vulnerable to the impact of global warming[1–6]. For breeding, stable land-fast sea ice (or fast ice), which is sea ice anchored to land, ice shelves, or grounded icebergs, is required[5,7]. The extent of the Antarctic fast ice is predicted to rapidly decline in the coming decades[8], and the species is predicted to lose 90% of its colonies by the end of this century. A significant decline of sea ice has been observed in 2023[9]. Despite this, the majority of emperor penguin colonies remain insufficiently studied due to the remoteness and harsh environmental conditions of their habitat[5,10]. New strategies to better understand this species and their responses to changing environmental conditions are urgently needed.

Of the 66 currently known emperor penguin breeding colonies[5,11], ground-truth population counts conducted during the winter,

at regular, frequent intervals, are only available for the colonies at Pointe Géologie in Adélie Land (average population size of 3900 breeding pairs)[12,13] and Atka Bay in Dronning Maud Land (average population size of 8600 breeding pairs)[14]. In addition, precise reproductive and life cycle parameters of the species have only been recorded from these two closely monitored colonies.

Emperor penguins return to their breeding colony at the onset of the dark Antarctic winter, between late March and early May, to begin their annual breeding cycle. After a period of courtship, copulation, and egg-laying lasting 6–10 weeks, females leave the colony to forage and replenish their body reserves at sea, while the males remain to incubate the single egg for an average of 64 days[15,16]. The birds limit their body heat loss during incubation by forming tight groups, so called huddles[15,16]. Chicks hatch in winter, i.e. between July and August,

[1]Department of Physics, Friedrich-Alexander Universität Erlangen-Nürnberg, Erlangen, Germany. [2]Department of Applied Ocean Physics and Engineering, Woods Hole Oceanographic Institution, Woods Hole, USA. [3]Université de Strasbourg, CNRS, IPHC UMR 7178, F-67000 Strasbourg, France. [4]CEFE, Université de Montpellier, CNRS, EPHE, IRD, Montpellier, France. [5]University of Moncton, Canada Research Chair in Polar and Boreal Ecology and Centre d'Études Nordiques, Department of Biology, Moncton, New Brunswick, Canada. [6]ENES Bioacoustics Research Laboratory, CRNL, CNRS, Inserm, University of Saint-Etienne, Saint-Etienne, France. [7]Institute of Biotechnology, University of Helsinki, Helsinki, Finland. [8]Centre Scientifique de Monaco, Département de Biologie Polaire, Monaco, Principality of Monaco. [9]These authors contributed equally: Céline Le Bohec, Daniel P. Zitterbart. ✉e-mail: alexander.winterl@fau.de; celine.le-bohec@cnrs.fr; dpz@whoi.edu

and are not thermally independent during the first 6–7 weeks of life, therefore one parent always stays with the chick while the other forages. After this brooding period, chicks are left alone at the colony, forming crèches with the other chicks, so that both parents can forage simultaneously to satisfy the chick's growing demands. Parents feed their chicks between 7 and 12 times until fledging between November and January[15,16], just before the land-fast sea ice begins to break up. The duration of these foraging trips declines from 15–29 days after hatching to <10 days before fledging[17–22].

Since the majority of colonies are not surveyed by ground-based observations, satellite-based surveys provide the bulk of available datasets for population size estimation. As the resolution of satellite imagery has improved over time, satellite-based surveys hold the greatest potential for estimating global populations and detecting trends[5,11,23–25]. However, even at the highest resolution currently feasible, which is on the order of 0.3 m/pixel, satellite-based surveys have not yet provided measurements at the individual level, but rather estimates of the number of animals based on the area occupied by the colony.

The viability of using colony area to estimate abundance suffers from uncertainties introduced by the imaging process (satellite off-nadir, sun azimuth, and sun elevation[23]), the conversion of colony areas to numbers of individuals, and large fluctuations of colony occupancy due to the species' phenology. The conversion of colony area to numbers of individuals requires knowledge of the average area number density (number of animals per square meter), which is subject to hourly fluctuations, as the animals regulate their body temperature by huddling together or loosening up, depending on weather conditions such as temperature, wind speed, solar radiation, and humidity[23,26,27]. Moreover, the occupancy fluctuations due to the annual phenology pattern is modulated by many factors such as sea ice extent and prey availability, which both influences the duration of foraging trips and the foraging success[20,28].

Due to the polar night, satellite images are not available during the incubation stage (June to July), when the least variation in the numbers of individuals is expected. Instead, usable satellite images can only be obtained between September and January, when chicks and only a fraction of the adults are present at the colony. Due to the low resolution of the images, it is not possible to distinguish between adults and chicks. For these reasons, satellite-based surveys suffer from large uncertainties when estimating the number of breeding pairs. Moreover, the breeding success of a colony based solely on the number of surviving chicks can not be determined[23,24]. Currently, satellite-based surveys may be sufficient to roughly gauge population size and long-term trends, but not to assess short-term selective forces impacting breeding success, unless the colony has completely disappeared[29].

The aim of this study is to develop a method to compensate for the uncertainties of satellite-based surveys and to provide an estimate of the annual number of breeding pairs as well as the annual breeding success of a colony, based on the colony area measured during the austral spring and summer (September to December). The method can be broken down into three separate steps. First, we convert colony covered areas from ground based or satellite imagery to individual counts by modeling the colony density as a function of temperature, wind speed, solar radiation, and humidity at the colony site ("windchill model"). Second, we present a phenological model that describes how the number of individuals present at the colony on each day depends on the number of breeders and the breeding success. We benchmark the model with ground based individual counts. Third, we invert the phenological model to infer the number of breeding pairs and the breeding success from sparse counts of adult animals at the site of the colony, obtained near the end of the breeding season. We benchmark this method with data from ground-based and satellite-based images.

## Results

### Converting colony area to colony size: windchill model

To obtain penguin counts from images in which the individual animals are not clearly distinguishable, we can multiply the measured colony area (ground area covered by penguins) with an estimate of the colony density (number of animals per area). For this, we use a so-called windchill model that predicts colony density from locally measured meteorological variables, specifically air temperature, wind speed, solar radiation, and relative humidity, as described in[27]. All of these meteorological variables affect the animals' heat balance and contribute to an apparent or perceived temperature, analogous to the windchill-corrected temperature reported in US-American weather forecasts. If the perceived temperature falls below a critical temperature, the animals tend to form huddles. Accordingly, the model requires a critical temperature as a further model parameter. Specifically, the critical temperature corresponds to the apparent temperature at which the average density of the animals within a colony reaches half of the maximum density, which we assume to be 12.8 animals per m² based on the maximum packing density for circles with 0.3 m diameter.

To estimate the windchill model parameters, we manually select the colony covered area from ground-based images of Pointe Géologie and Atka Bay[14,30] using the image annotation software Clickpoints[31], project the resulting polygons to the top view projection[32], and calculate the total occupied area in m² (Fig. 1A, B). From this, we compile a dataset of 538 measurements of colony area, number of individuals (chicks and adults), and meteorological variables (temperature, wind speed, solar radiation, and humidity) acquired at each colony between September and December of three (AB: 2018, 2019, 2020) and four (PG: 2014, 2015, 2016, 2017) seasons. From the correlation of colony density fluctuations (colony area divided by the number of individuals) with fluctuations in meteorological variables, we estimate the windchill model parameters (Fig. 1C–F, Supplementary Fig. 5, Supplementary Table 3).

The windchill model predicts the measured colony density with an $R^2$ value of 0.32 and an average geometric error of 39%. This error may seem large, but when we multiply the density predicted from the model with the measured area in order to derive the number of individuals (see Supplementary Note 6), this matches the actual animal counts with an $R^2$ value of 0.93 and an average geometric error of 11%. Moreover, we find that 60% of the individual count data fall within the predicted 1-sigma interval (Fig. 6F). Figure 1H, I show an example of large fluctuations in measured colony area, and measured as well as model-predicted number of individuals at Pointe Géologie between September 1 and December 31 of 2014.

Accordingly, we find large fluctuations of colony density over the course of a day (e.g. from 0.20 to 2.41 animals per m² on 2014-10-08, with a total range between 0.03 and 7.00 animals per m² during the whole observation period; Fig. 1G). These large density fluctuations explain the large uncertainties of satellite-based counts when a constant density of 0.93 animals per m²[24] is assumed. Estimating breeding success from adult penguin counts: phenological model

### Estimating breeding success from adult penguin counts: phenological model

We count the number of emperor penguin adults and chicks on a weekly basis at Atka Bay (AB, 70° 40′S, 8° 16′W) over 3 breeding seasons (2018–2020), and at Pointe Géologie (PG, 66° 40′S, 140° 01′E) over 10 breeding seasons (2012–2021). The data reveal a characteristic pattern of animal counts depending on annual cycles of incubation, guarding, foraging at sea, and feeding chicks at the colony (Fig. 2).

Based on known patterns of the species' breeding activity[17–22], we develop a mechanistic phenological model that describes the observed colony abundance during a typical breeding season (from March 1 to February 28 of the next year). This model describes the

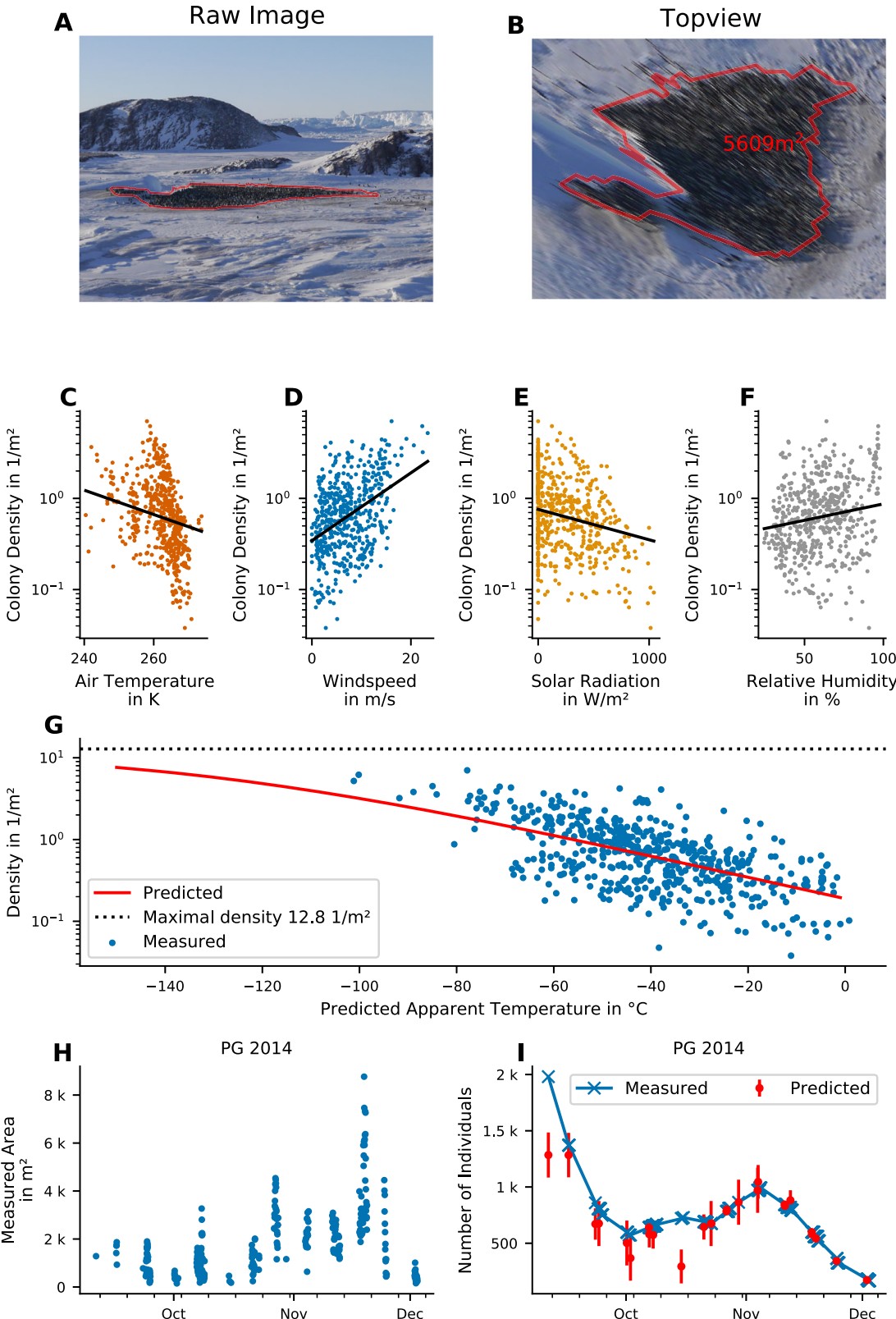

presence/absence pattern for a breeding pair and its chick based on the following set of parameters: time of arrival at the colony site, courtship duration, first female absence duration after laying, foraging trip duration, and period at the colony to feed the chick. The model then calculates the total number of females, males, and chicks within the colony at any given time point from the presence/absence patterns. We fit the model parameters to the observed number of adults,

using a Markov chain Monte Carlo approach, which iteratively optimizes the parameters and returns a distribution for each model parameter as well as a distribution for the number of chicks and adults for each day of the breeding season.

To validate the model, we compare the observed numbers of individuals to the model predictions (see Fig. 3). Because the count data usually vary on a logarithmic scale, we report not arithmetic,

**Fig. 1 | Predicting colony density with the windchill model. A** Example of a ground based image recorded on 2017-08-22 04:00:00 UTC at Pointe Géologie. The manually-annotated outline of the colony is highlighted with a red polygon. **B** Projected top view of the image shown in (**A**). The number indicates the area covered by the colony. **C–F** Correlation of measured colony density with the meteorological values: air temperature (**C**), windspeed (**D**), solar radiation (**E**), and humidity (**F**). The y-axis shows the colony density in penguins per square meter, the x-axes show the respective meteorological variables. Each dot represents one image. The black lines show the corresponding log-linear regression line. The slopes correspond to the model parameters. **G** Dependence of measured colony density on apparent temperature $T_a$. Each dot represents the data from one image

and time point. The red line shows the model prediction (fit of the sigmoidal function (Eq. 2) to the data). **H** Surface area covered by the colony (in square meters) for Pointe Géologie between September 1 and December 31 in 2014 over time, estimated from ground-based images. The observed short-term variance (seen as vertical stacks in the data points) are due to daily variation in colony area, driven by environmental parameters. **I** Measured (blue crosses) and predicted (red dots) animal count over time. Predictions are based on the measured areas shown in (**H**) for Pointe Géologie between September 1 and December 31 in 2014 multiplied with the density as predicted by the windchill model. Error bars of the predicted values are standard deviations of multiple images per day ($n = 285$ images over 26 days). Source data are provided as a Source Data file.

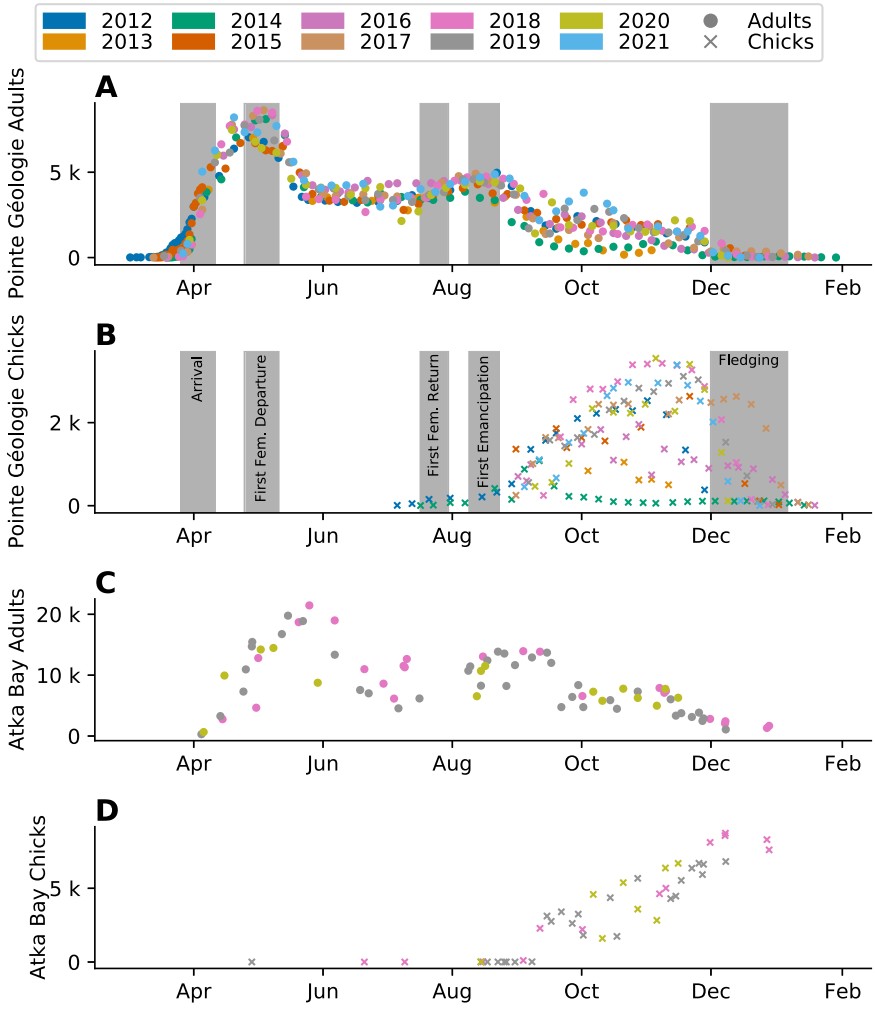

**Fig. 2 | Manual counts of the number of individuals.** Number of individuals at Pointe Géologie (**A**, **B**) and Atka Bay (**C**, **D**) colonies. Circles indicate counts of individual adults (**A**, **C**), crosses indicate counts of individual chicks (**B**, **D**). Colors denote different breeding seasons. Gray bars show the time range of key phenological events extracted from manual observations at Pointe Géologie (**A**, **B**). Note that the number of chicks increases between August and November as the chicks become thermally independent and therefore less visually obstructed. Source data are provided as a Source Data file.

but geometric errors: for example, a ± 25% geometric error corresponds to a 1.25-fold overestimate (multiply by 1.25) or a 1.25-fold underestimate (divide by 1.25). We find an average geometric error of 16% for all data points (AB: 23%, PG: 15%). 81% of the observations fall within one standard deviation around the model estimate (AB: 83%, PG 81%). The coefficient of determination ($R^2$ value) between model and data is 0.91 ($n = 518$ counts, $p < 0.01$) for all data points (AB: $R^2 = 0.73$, $n = 74$ counts, $p < 0.01$, PG: $R^2 = 0.95$, $n = 444$ counts, $p < 0.01$).

A key feature of our model is its ability to predict the annual breeding success expressed as the number of fledging chicks relative to the annual number of breeding pairs. Because we only estimate the

model parameter values based on the number of adults, we can assess the predictive power of the model by comparing the number of chicks derived from the model with the actual number of chicks observed. We find an overall $R^2$ value of 0.45, with $R^2 = 0.64$ for Pointe Géologie and $R^2 = 0.12$ for Atka Bay (Fig. 3). The average geometric error is 41% for all data, with 51% for AB data and 39% for PG data. The large relative error for Atka Bay is attributable to the unusually (compared to other AB and PG seasons) high number of chicks that were still present late in the 2018 season (see Fig. 3K), which the model fails to predict. Further, the average number of sample counts per season for Atka Bay (26 counts) is lower than for Pointe Géologie (44 counts). In addition to the

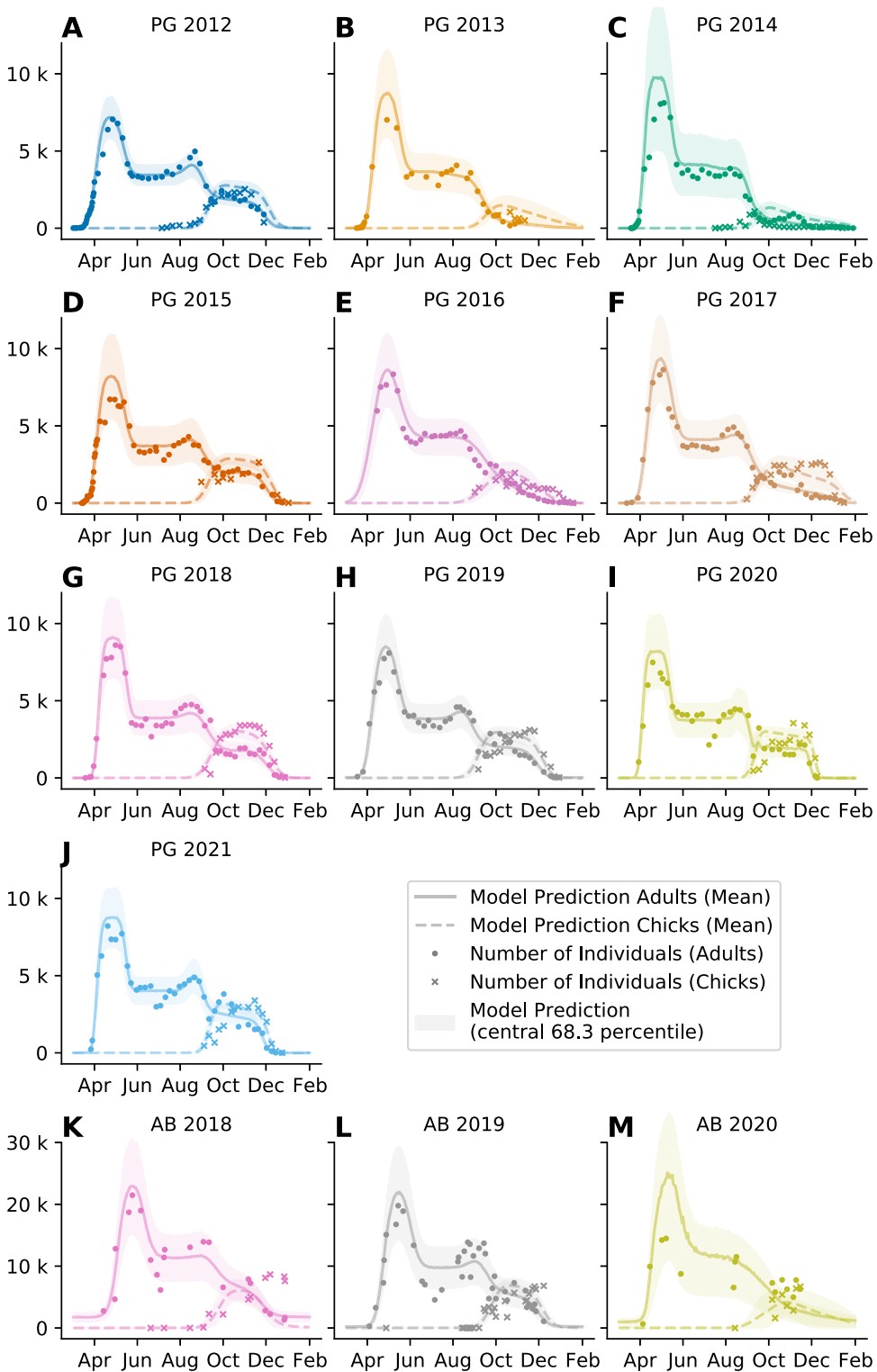

**Fig. 3 | Phenological model fit to manual counts.** Overlay of data and model output for all observed seasons at Pointe Géologie (10, **A**–**J**) and Atka Bay (3, **K**–**M**) colonies. Individual panel titles indicate colony (PG Pointe Géologie, AB Atka Bay) and season. *Y*-axes show the number of individual adults (circles) and chicks (crosses). Solid and dashed lines show the mean model prediction for the adult and chick counts, respectively. The shaded areas indicate the ± sigma confidence interval of the model prediction. Colors indicate the season as in Figs. 2, 4 and 8. Source data are provided as a Source Data file.

number of alive adults and chicks per day, the total number of fledging chicks, dead chicks and lost eggs are available from Pointe Géologie but not from Atka Bay. For Pointe Géologie, the model predicts the number of fledging chicks and dead chicks with R² values of 0.74 and 0.32, and average geometric errors of 25% and 56%, respectively (Fig. 4A, B). The model predicts the number of losteggs with a mean

absolute error of +/−190 eggs and average geometric errors of 28%. The model also provides date estimates for phenological events such as arrival, first departure and first return of the females after laying, chick emancipation (when the chick is thermally independent to join the other chicks in crèches), and fledging (when the chick has molted and leaves the colony for the first time). We compare these estimates

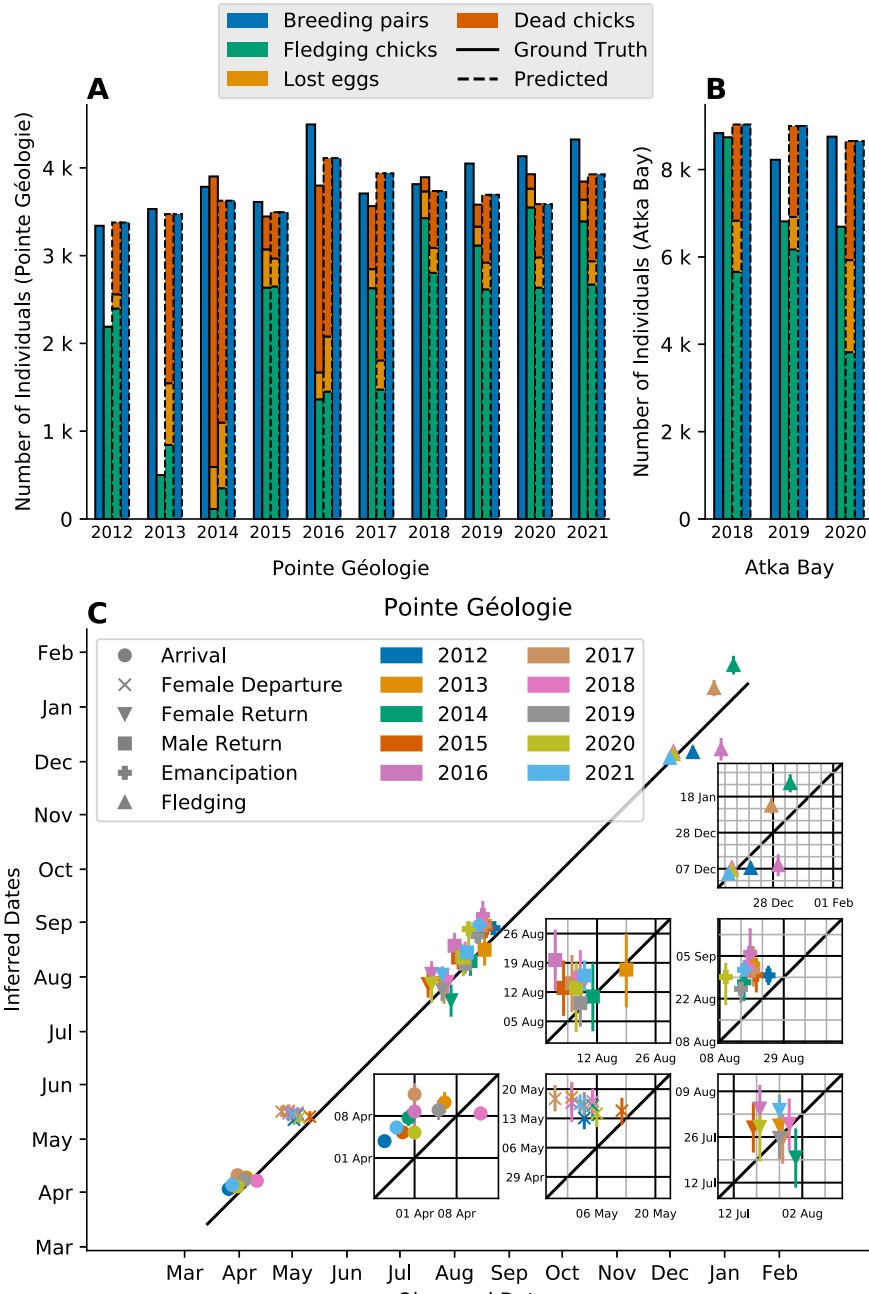

**Fig. 4 | Model estimates of key phenological parameters. A, B** Observed and predicted number of breeding pairs (blue), lost eggs (orange), dead chicks (red), and fledging chicks (green) by year and colony. For each year, the two left bars with a solid line show the observed numbers, and the two right bars show the predicted numbers. For each year, the outer bars show the number of breeding pairs, the inner bars show the breeding result splitted in three stacked sections: fledging chick (bottom), lost egg (middle), and dead chick (top). The predicted numbers of lost eggs, dead chicks, and fledging chicks add up to the number of breeding pairs by model definition, while the ground-truth values do not, due to counting inaccuracies. **C** Time of predicted phenological events vs observed phenological events. Each point corresponds to one season; the y-bars show the 1-sigma confidence interval around the mean value ($n = 13200$ MCMC-samples). Colors indicate the seasons. The black line shows the line of identity. Insets show the same data, zoomed in for better visibility, to show inter-annual variation. Gridlines indicate weeks. Note that systematic shifts between observed and estimated dates arise from the difference of first occurrence (manual observations) and central event date (model estimates). Source data are provided as a Source Data file.

with ground-based observations for Pointe Géologie (Fig. 4C, Supplementary Table 2). The timing of events as estimated by the model shows good agreement with the observed timing of events (average absolute error of 10 days) when pooled over all years. For individual years, we find good agreement of the model prediction for all events. Note that a difference of 14 days arises between model prediction and ground-truth times for female first departure and chick emancipation, because these events are recorded in the field as the time of the first observation of the respective behavior, whereas the model predicts the central moment of the event. The model describes each season and each colony separately. Therefore, we estimate 13 sets (from 13 breeding seasons) of 14 parameters ($BP$, $H$, $F$, $NB$, $t_0$, $\Delta t_0$, $m$, $b$, $\Delta b$, $c_{max}$, $c_{min}$, $s_{max}$, $s_{fem}$, $s_{min}$) that provide insight into the interannual and inter-colony variation of the phenology (Supplementary Fig. 1).

Most prominently, the arrival time ($t_0$) at Pointe Géologie (April 7 +/− 3 days) is significantly (MWU: Mann-Whitney-U-Test, statistic =

## Table 1 | Phenological Model Parameters

| Symbol | Name | Lower Limit | Upper Limit | Unit | Description |
|---|---|---|---|---|---|
| $t_0$ | arrival date | 50 (Feb 20) | 150 (May 30) | days | Peak of animal arrivals at the colony |
| $\Delta t_0$ | arrival date standard deviation | 4 | 14 | days | Standard deviation of arrival dates |
| $m$ | courtship duration | 28 | 42 | days | Average duration between arrival and first departure of females |
| $b$ | female absence duration | 50 | 100 | days | Average duration between first departure and return of females |
| $\Delta b$ | female absence duration standard deviation | 0 | 14 | days | Standard deviation of duration of female absence |
| $BP$ | number of pairs | 2.000 | 15.000 | pairs | Number of pairs that mate and produce an egg |
| $NB$ | non breeder to breeding pair ratio | 0 | 1 | 1 | Ratio of birds that do not find a breeding partner and leave the colony before the breeding period, to the total number of breeding pairs |
| $H$ | hatching success ratio | 0 | 1 | 1 | Ratio of pairs that repeatedly return after the breeding period, relative to the number of breeding pairs at the beginning of the breeding season |
| $F$ | fledging success ratio | 0 | 1 | 1 | Ratio of pairs that repeatedly return until the end of the chick rearing phase, relative to the number of breeding pairs at the beginning of the breeding season |
| $c_{max}$ | maximum time at colony per trip | 1d | 5d | days | Time one parent spends at the colony per foraging trip during the guarding phase and at their first foraging trip |
| $c_{min}$ | minimum time at colony per trip | 1d | $c_{max}$ | days | Time one parent spends at the colony on their last foraging trip |
| $s_{max}$ | maximum time at sea per trip | $c_{max}$ | 21d | days | Time one parent spends away from the colony per foraging trip during guarding phase and at their first trip after chick emancipation. This includes foraging and commuting time |
| $s_{min}$ | minimum time at sea per trip | $c_{max}$ | $s_{max}$ | days | Time one parent spends away from the colony per trip on their last foraging trip. This includes foraging and commuting time |
| $s_{fem}$ | time at sea at first trip (only females) | $c_{max}$ | $s_{max}$ | days | Duration of the first foraging trip of females after hatching. It is shorter or equal than the following trips |

Parameter names, descriptions, and numerical ranges. Note that all parameters are estimated by the model when fitted to the manual counts. However, when applied to satellite based counts with as little as 6 data points per season, only BP and F are estimated, ensuring a robust fit. $t_0$ is inferred from our observation of latitudinal shift. All other parameters are fixed to their mean values as estimated from the manual counts.

0.0, $n = 13$ seasons, $P < 0.01$) earlier than at Atka Bay (April 27 +/− 7 days). The number of breeding pairs ($BP$) also shows significant ($MWU = 0.0$, $n = 13$ seasons, $P < 0.01$) differences between those two colonies (AB: 8600 breeding pairs on average, PG: 3900 breeding pairs on average). In addition, we find statistically significant (see Supplementary Note 2) interannual variance for the number of breeding pairs ($BP$, AB: +/− 620), fledging success ($F$, PG: +/− 0.66), time of arrival ($t_0$, AB: +/−12.0 days, PG: +/−4.6 days), width of arrival date distribution ($\Delta t_0$, PG: +/− 3.8 days), width of female return date distribution ($\Delta b$, PG: +/− 7.0 days), minimum time at sea ($s_{min}$, PG: +/−6.4 days), and maximum time at sea ($s_{max}$, +/− 6.7 days). Note, however, that the standard deviation for the duration of each event is smaller than the sampling interval of 7 days for the ground-truth data. Consequently, the biological significance of these interannual variations cannot be reliably verified.

We further investigate the correlation between model parameters versus breeding success, as estimated by the ground-truth ratio of fledging chicks to breeding pairs (see Supplementary Fig. 3). We find significant ($P < 0.02$, $n = 13$ seasons) correlations between breeding success and the foraging trip duration during the crèching period with the following $R^2$ values: 0.54 ($s_{max}$), 0.52 ($s_{min}$), 0.46 ($c_{min}$) (see Table 1, Supplementary Fig. 2). We then sum the duration of all foraging/feeding trips to obtain the total time spent at sea or at the colony during the crèching phase (Fig. 5). We find a significant ($P < 0.03$, $n = 13$ seasons) correlation between breeding success and total time spent at sea ($R^2 = 0.55$) or time spent at the colony ($R^2 = 0.39$). Noteworthy, a longer total time spent at the colony correlates with a higher breeding success, while a longer total time spent at sea correlates with a lower breeding success.

## Applying the windchill and phenological models to satellite imagery

To showcase an application, we use previously published data of colony area from the population of Coulman Island (CI), Atka Bay (AB), and Stancomb-Wills Glacier (SW) in 2011[23]. We obtain the local meteorological data at the time of satellite image acquisition from[33] and, based on the windchill model, predict a colony density between 0.6 animals/m² to 1.8 animals/m². After multiplying the colony area with the colony density, we obtain animal counts, which are the sum of chicks and adults present at the colony. Note that if multiple images are recorded on a single day, the predicted counts are averaged in order to reduce sampling bias. Finally, we fit the phenological model to the counts to derive the number of breeding pairs and the number of fledging chicks for each year. The results are summarized in Supplementary Table 4. For these three colonies, low quality ground-truth data for the number of breeding pairs exist, albeit not from 2011 but from earlier years[24]. Nonetheless we find an average geometric error of 28% and an R² score of 0.88.

We also benchmark this approach using ground-based images from AB and PG. The procedure is analogous: we obtain the colony area from the images, use the windchill model to estimate colony density based on meteorological data from station observatories, multiply both numbers to derive the number of animals, fit the phenological model to the counts, and arrive at a prediction for the number of breeding pairs and fledging chicks for each year that we can compare with ground-truth counts. We find good agreement, with an average geometric error of 21%, an R² value of 0.92, and 74% of the ground-truth data being within one standard deviation around the model prediction (Fig. 6A–F).

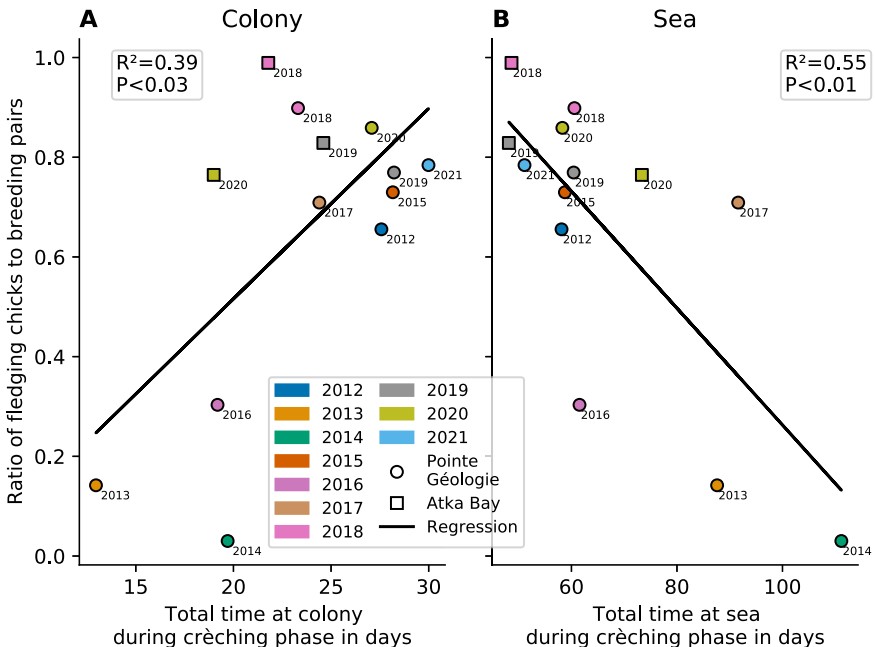

**Fig. 5 | Correlation between foraging pattern and breeding success.** The x-axis shows the total number of days spent at the colony (**A**) or at sea (**B**) during the crèching phase as predicted by the model, averaged for females and males. The y-axis shows the ratio of fledging chicks to breeding pairs (breeding success) from manual observations. Every dot (Pointe Géologie) and square (Atka Bay) denotes one season (color and labels show corresponding season). The black lines show the regression lines for each point cloud (Pointe Géologie and Atka Bay merged). We find a significant (*P* = 0.023) positive correlation between breeding success and time at colony, and a significant (*P* = 0.004) negative correlation between breeding success and time at sea. Both hypotheses are tested with a two-sided Wald test from *n* = 13 seasons and without multi-comparison correction. Source data are provided as a Source Data file.

## Discussion

### Phenological model

We present a mechanistic model that estimates phenological information from incomplete time series of animal counts in emperor penguin colonies. The phenological information predicted by the model include time of first arrival at the colony site to breed, number of breeding pairs, number of hatched chicks, number of fledged chicks, timing of foraging trips, duration of courtship, and female absence (during incubation). Our phenological model is able to recapitulate the temporal fluctuations of the weekly individual counts at two colonies (Atka Bay and Pointe Géologie) over several seasons, and it predicts the number of chicks per day and the number of fledging chicks solely based on the counts of adult animals.

Predicted and true counts of chicks and adults per sampling day align less well at Atka Bay than at Pointe Géologie, probably due to a lower number of sample counts for Atka Bay than for Pointe Géologie (26 vs 44 counts per season). Alternatively the phenology at Atka Bay might be more variable than at Pointe Géologie. Studies at other colonies or further data acquisition at Atka Bay could provide further insights on this issue. Our model is therefore robust in detecting the main features of the phenological pattern, while struggling with very fine predictions like weekly counts, when the underlying data are sparse. We attribute this to the probabilistic inference method that blurs the parameter space when confronted with noisy and sparse data.

The model has difficulty in years with highly unusual presence/absence patterns at specific times in the breeding cycle. For example, in 2017 at Pointe Géologie, the number of adults present during the chick rearing period was much lower than the number of chicks. The model (which is informed only about the number of adults but not the number of chicks) therefore estimated a survival rate of <50%, while in reality, significantly more chicks survived (71%). We hypothesize that, in 2017, adults spent extremely long periods outside the colony, reflecting an extended sea ice cover at the end of chick rearing[34], while

food resources were likely available and abundant near the sea ice edge, enabling them to feed their chick sufficiently until they fledge. Moreover, the model is based on very limited data on the individual trip durations during the chick-rearing phase (i.e. the study of Kirkwood & Robertson 1997[19]), which was carried out over a single year on the Auster and Taylor Glacier and Auster colonies. To refine the model, it would be important to collect foraging information on this critical period of the breeding cycle, over several years and on several colonies facing contrasting environmental conditions.

Overall, the model performs well at predicting the number of fledging chicks and breeding pairs, although it predicts larger numbers of dead chicks and lost eggs compared to manual counts (Fig. 4A). A part of the discrepancy arises because lost eggs and dead chicks are counted only if they can be found, but are often covered by snow. Therefore, the model may in fact provide more reliable predictions for the number of dead chicks and lost eggs in years with typical phenology.

How well the model performs the timing of phenological events is difficult to assess because ground-based observations are only intermittent and moreover based on single events of individual animals, for example the first penguin arriving at the breeding site. By contrast, the phenological model takes all available counts into consideration and from that computes the central tendency of events, e.g. the peak arrival time. The temporal agreement between model predictions and observations are within the sampling interval of about 1 week for ground-based observations, plus the timing offset between the first occurrence and the event peak (see Fig. 4C insets, S 11). We tested our model against a second colony location (Atka Bay) and found a similarly good agreement, despite the model being mainly built on behavioral knowledge from Pointe Géologie. Nonetheless, additional ground-truth data from other colonies will help confirm the generalizability of the model.

We could not quantify how well the model can predict annual variations in phenological events at the same colony, because most

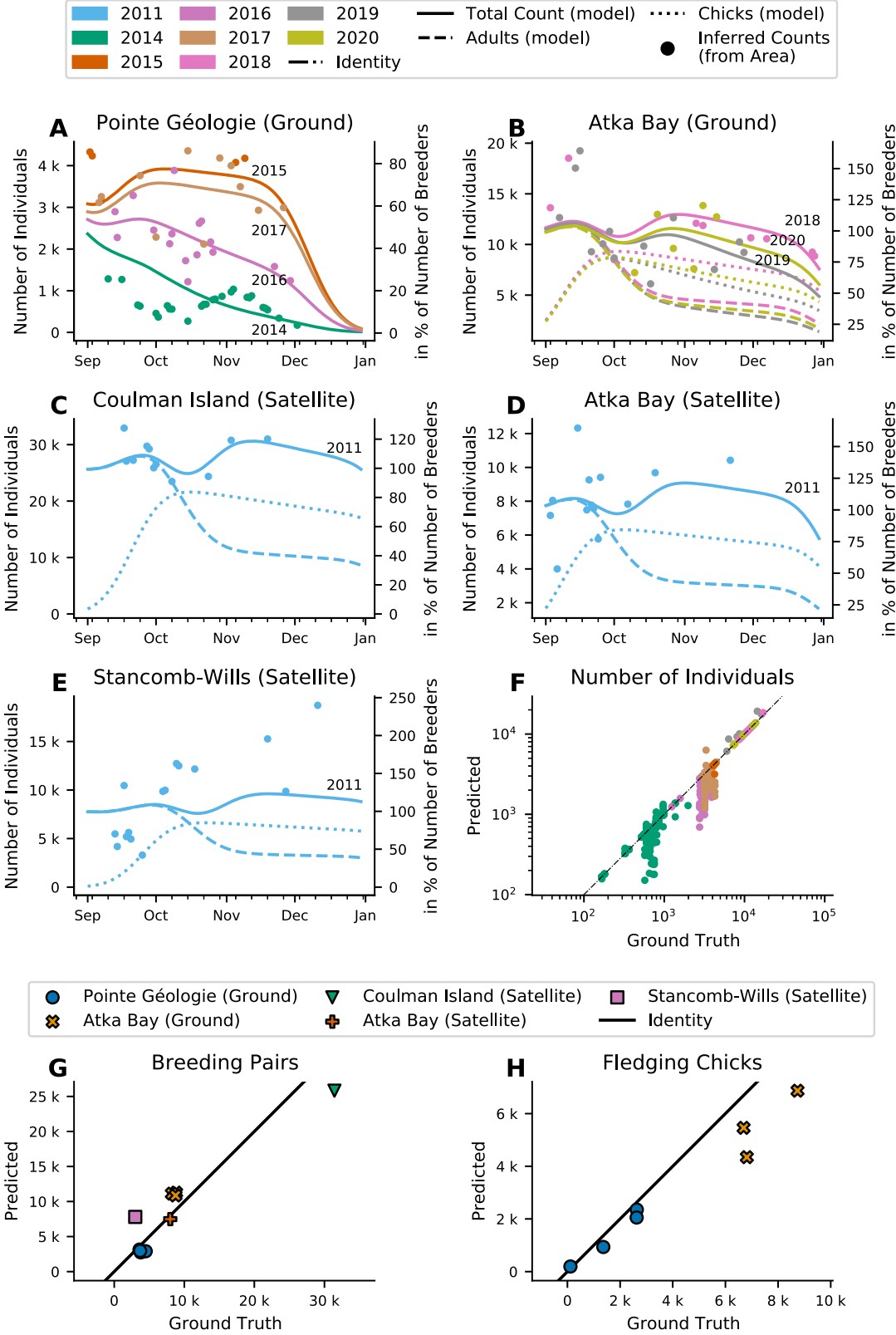

**Fig. 6 | Application to satellite data. A–E** Fit of the phenological model to individual counts inferred from measured colony area and the windchill model predictions (dots). Lines show the phenological model predictions for adults (dashed), chicks (dotted), and total counts (solid). In (**A**), adult and chick counts are omitted for readability. Those data are provided in Supplementary Fig. 4. Colors and labels (total counts, solid lines) denote seasons. **F** Comparison of ground-truth counts and counts inferred from colony area and windchill model predictions for Pointe Géologie and Atka Bay over 7 seasons as denoted by colors. Note: Pointe Géologie data covers 2012–2017, Atka Bay 2018–2020, therefore colors are also colony specific. The back line shows line of identity. **G, H** Comparison of ground-truth and phenological model predictions for the number of breeding pairs (**G**) and fledging chicks (**H**). Ground-truth values for Pointe Géologie and Atka Bay (Ground) are from manual counts, all others are from ref. 24. Note that for satellite data, there is no ground-truth for the number of fledging chicks. Colors and marker shape denote season. Source data are provided as a Source Data file.

events vary less than the resolution of ground-based observations (7 days, see Fig. 4C). Nonetheless, some interesting model predictions emerge. For example, the duration of courtship ($m$) and the females' first absence from the colony after laying ($b$) (the incubation period for the males), do not show significant (MWU, p>>0.05) variability between years and colonies, which is to be expected because such processes have likely evolved to be tightly regulated at a physiological level specific to the species. We also found a negative correlation of the total time spent at sea with breeding success: long (>10 days) foraging trips at the end of the chick-rearing phase (between September to December) lead to lower (<50%) breeding success, likely because the frequency of feeding events decreases, exposing the chick to a higher risk of starvation in between feeding events. We suggest that increased foraging trip durations are linked to longer distances to the land-fast ice edge, which have been shown to negatively impact fledging success[34].

The model-predicted as well as measured arrival times ($t_O$) and first hatchings ($t_H = t_O + m + b$) between Pointe Géologie and Atka Bay differ by ~30 days. At Pointe Géologie, the more northerly colony (66.39°S), first hatchings are observed in the 1st week of July, while at Atka Bay (70.60°S), hatchings are estimated (by our model) to occur in the first 2 weeks of August. In both colonies, the timing of hatchings correlates with the first sunrise after mid winter (June 29 at Pointe Géologie and July 28 at Atka Bay, Supplementary Fig. 6, Supplementary Note 4). A positive correlation between arrival times to breed and latitudes has been previously reported[35]. A likely explanation of this correlation is that penguins synchronize the chick rearing phase and therefore their whole annual cycle with the increasing abundance of prey. Prey abundance, in turn, is triggered by a rise in primary production due to an increase in solar radiation and/or a decrease in sea ice cover near the Antarctic continent. A similar synchronization between prey abundance and breeding cycle has been observed in numerous species, including other seabirds[36], and is in line with the match/mismatch hypothesis[37–39] that states that the reproductive success of a species depends on its ability to match its phenology with that of its prey species. This relationship between behavior and latitude helps to increase the precision of our phenological model when animal count data are too sparse to reliably estimate event times such as arrival times. However, more data on the arrival time of breeders at colonies of different latitudes are necessary to consolidate this hypothesis.

## Windchill model

To protect against the cold, emperor penguins form tight huddles, which implies a relationship between colony density and meteorological conditions. This relationship can be described by a previously reported model (which for simplicity we refer to as the windchill model)[27]. We re-trained the windchill model with data from Atka Bay and Pointe Géologie obtained between September and December, as the previously reported model parameters were trained with data obtained during the high-density (up to 15 animals per m²) courtship and incubation phase of the austral winter, in April and May.

The re-trained model can partially explain the fluctuations in colony density with a correlation of $R^2 = 0.32$ and 60% of data within the predicted 1 sigma interval, which is slightly higher than the predictive power of the original model (50%[14]). A less-than-perfect predictive power is to be expected because of the mild weather conditions of the austral spring, and because a mixed colony of adults and chicks rarely adopts a configuration where the majority of the animals are found in dense huddles. Furthermore, local weather conditions can be subject to considerable uncertainty, either because they are extrapolated from data recorded at the nearest weather station (as for the Pointe Géologie and Atka Bay (2018–2020) estimates), or because they are derived from weather models (as for the Stancomb Wills, Coulman Island, and Atka Bay (2011) estimates).

The huddling behavior of emperor penguins and hence the colony density not only depends on weather conditions but also on the thickness of the animals' insulating fat layer. Therefore, the parameters of the windchill model will change over the course of the breeding season as the males lose body fat during their incubation time, or a s well-fed females return to the colony. Moreover, seasons with poor food availability will result in lower individual fat reserves, which would also be reflected in altered model parameters. Currently, we have not explored the relationship between fat layer thickness and changes in windchill model parameters. The single set of parameters available today that we used in our study can predict colony density at the end of the breeding season for the Atka Bay and Pointe Géologie colonies, but may not be fully representative for other times in the breeding season, and possibly also not for different colonies.

## Estimating abundance and breeding success from remote sensing

Optical satellite images, currently used to estimate populations, can only be taken during periods with sufficient light conditions when occupancy is most erratic, which makes it challenging for inferring population size and breeding success. One of the main objectives of our phenological model is to provide more reliable estimates of the number of breeding pairs within a colony from those images. Currently, it is common practice to multiply the colony area A (in units of m²) as measured from satellite images with a conversion factor CF of 0.93 breeding pairs per m²[24]. However, depending on the weather conditions and the time of year, the colony area and animal density fluctuate, potentially contributing to large uncertainties[23,24]. To reduce these uncertainties, we suggest the following approach for a reliable estimate of the number of breeding pairs: (1) Collect several (5–10) satellite images from multiple time points t (over a time period of 2 months, from October to December) and measure the colony area A; (2) Acquire local weather conditions at the time of satellite image capture and estimate colony density $\varrho$ using the windchill model; (3) Compute the total number N of animals in the colony ($N = \varrho$ A); (4) Fit the phenological model to N(t) and extract the number of breeding pairs BP and fledged chicks F.

To benchmark this method, we count the number of breeding pairs from ground-based images of the colony (instead of satellite images) for our two colonies (AB and PG) over a time period of 3 and 4 years, and estimate the number of breeding pairs from satellite images using our model. We find an excellent correlation ($R^2 = 0.93$) between ground-truth and estimated numbers. Furthermore, we compute a conversion factor CF from the number of breeding pairs and the colony area as $CF = BP/A$ (see Supplementary Note 5). We obtain a value of $CF = 1.09$ breeding pairs per m², which is close to the conversion factor of 0.93 breeding pairs per m² as reported in[24]. Note that when we compute the average CF, we exclude an outlier from the 2014 season at Pointe Géologie with an unusually low (<5%) breeding success, a very low number of animals in the colony (~500 instead of the usual 1800 animals on average), and a very high density of 5.98 breeding pairs per m². However, this outlier is not excluded when we compute the correlation of the model estimates with manual counts. Thus, previous estimates of the total number of breeding pairs from satellite images are accurate on average, but individual counts from any given image can be subject to large errors, as was already pointed out in[23,24]. The number of surviving chicks that we estimate with our model following the approach outlined above also agrees well with manual counts ($R^2 = 0.81$ and average geometric error of 26% based on data from two colonies over 3 or 4 years).

Our data show a striking difference in the number of breeding pairs at Atka Bay between the periods 2008–2011 and 2018–2020. In 2008–2011, the estimates range from 7300 (satellite images, Supplementary Table 4) to 9657 (satellite images[24,]), while in 2018–2020, the

estimates are around 10,000 (both from ground-truth counts and ground-based images). This increase by >2000 breeding pairs is likely due to immigration, recruitment or occasional visits of individuals from close by colonies. A possible reason for migration of individuals is the neighboring Halley colony that experienced three consecutive years of breeding failure (from 2014 – 2016) and was completely abandoned in 2016[24].

Recent satellite based publications[6,23,24] as well as this study report rather high discrepancies in the estimated colony population sizes compared to historical aerial and ground based surveys (i.e. Atka Bay −700 animals, Stancomb Wills + 3400 animals, Coulman Island −5832 animals, see Supplementary Table 5). We account these differences to improvements in the estimation of the number of breeding pairs and not to erroneous counts (on either side). Historical ground counts did not account for the phenological status of the colony and simply reported the number of individuals in the colony at a given time[24], while modern satellite-based surveys (and this study) aim to estimate the number of breeders. See Supplementary Table 5 for a comparison of different censuses.

While a reliable number of breeding pairs provides valuable information about the current status of the species, abundance information needs to be collected over many decades to use it as a population predictor in a long-lived species[23]. Similarly, other recent work[40] cannot clearly determine population trends from the number of breeding pairs alone, even when considering decades of satellite-based observation. This could possibly be due to the lack of a phenological correction to their data. In contrast, the number of surviving chicks during a breeding period is a much more sensitive predictor of future population trends, e.g. due to global change or changes in food supply, especially if collected annually. As of now, with our presented method, we have not detected a declining trend in the number of surviving chicks at AB or PG over the 2012–2021 period. However, given the recent development of a dramatically decreased sea ice extent around Antarctica[9,41], the Emperor penguin will face many challenges for continued successful breeding, and close monitoring of the species is becoming imperative.

Our next milestone will be to apply this method on a long-term circum-Antarctic scale, enabling us to use the breeding success of emperor penguins as an early warning indicator, very much like the canary in a coal mine, as an early warning indicator for the Southern Ocean ecosystem.

Future satellites will be monitoring earth at very high resolution and revisit rates at lower costs, while automated image detection techniques will drastically simplify the analysis, rendering our process feasible.

If applied to a circum-Antarctic scale, our method has the potential to alert the scientific community annually about each colony's breeding success anomalies and enable us to better pinpoint their causes in correlation with local and global oceanographic and climatic events. Ultimately, we aim to provide stakeholders and governments with information on the health of animal populations and their ecosystems in the Southern Ocean, to rapidly implement effective conservation measures and to monitor their success.

## Methods
### Animal ethics statement
All studies providing data for this study were conducted at two Emperor penguin colonies (Atka Bay and Pointe Géologie) on the Antarctic continent. The studies were approved by the environmental agencies and ethics committees responsible for the respective regions.

Atka Bay: All procedures were approved by the German Environment Agency (Umweltbundesamt-UBA), permit no.: II 2.8−94033/100, II 2.8−94033/166, II 2.3 − 94032/1.

Point Géologie: French ethics committee (APAFIS#29338-2020070210516365 and APAFIS#4897-2015110911016428) and the French Polar Environmental Committee of the Terres Australes et Antarctiques Françaises (TAAF project implementation and access permits 2012-117 & 2012-126, 2013-74 & 2013-82, 2014-116 & 2014-132, 2015-52 & 2015-105, 2016-76 & 2016-82, 2017-92 & 2017-102, 2018-116 & 2018-129, 2019-107 & 2019-115, 2020-65 & 2020-72, 2021-40 & 2021-51 & 2021-102).

### Colony area and colony density: windchill model
While they incubate their egg during the harsh Antarctic winter, emperor penguins conserve energy by forming tight groups, the so-called huddles[26,42]. The fraction of individuals of a colony that are currently in a huddle changes depending on an apparent (i.e. subjectively perceived) temperature. The apparent temperature depends on four environmental variables: ambient temperature (T), windspeed (W), solar radiation (R), and humidity (H). These variables linearly contribute to the apparent temperature $T_a$, with linear factors $c_W$ (windchill factor), $c_R$ (solar radiation factor), $c_H$ (humidity factor), and a factor of unity for ambient temperature (Eq. 1)[27].

The model then describes the colony density as a sigmoidal function of the apparent temperature (Eq. 2). For very low apparent temperatures, the colony density tends to a maximum value of 12.8 animals / m². This maximum density corresponds to a hexagonal packing of cylinders with a diameter of 30 cm. At very high apparent temperatures, the colony density tends to zero. The sigmoidal function has two free parameters, the critical temperature Tc at which the density is half of its maximum, and the steepness value $b_0$ that describes how sensitively the animals respond to temperature changes.

$$T_a = T + c_W W + c_R R + c_H H \qquad (1)$$

$$N/A = \rho = 12.8/(1 + \exp((T_a - T_c)/b_0)) \qquad (2)$$

We estimate the model parameters $b_0$, $c_W$, $c_H$, $c_R$, and $T_c$ from the density data recorded at Pointe Géologie and Atka Bay between 2014 and 2020 for the months of September to December using Markov chain Monte Carlo sampling (see Supplementary Note 1). We can then predict the density of any colony at any given time point between September and December from weather data alone.

Colony area was measured from images after perspective correction[32]. The Pointe Géologie colony was monitored between 2014 and 2017 using automatic time lapse cameras[30]. For Atka Bay, we used the panoramic images from the seasons 2018, 2019, and 2020. We manually marked the colony boundaries on 538 images (509 of Pointe Géologie, 29 of Atka Bay) acquired between 1st of September and 31st of December of each season. An example of an annotated image is shown in Fig. 1A. From the colony boundaries we computed the surface area covered by the colony (in m²) by perspective correction and projection using the intrinsic and extrinsic camera parameters such as focal length, elevation, tilt, and roll[32]. A projected top view is shown in Fig. 1B. The extracted areas of the Pointe Géologie colony between September 1 and December 31, 2014 are shown in Fig. 1H.

Colony densities (average number of individuals (adults plus chicks) per area) for Atka Bay and Pointe Géologie are calculated from individual counts (weekly resolution) divided by colony area for each of the images. For time points where we had area measurements but not a corresponding individual count, we linearly interpolated the value from the two nearest available counts.

We extended our dataset of colony area with satellite image based measurements of colony area from Coulman Island (12 days), Atka Bay (19 days), and Stancomb-Wills Glacier (18 days), between September 10 and December 11, 2011 published in ref. 23.

The meteorological data for Pointe Géologie and Atka Bay (seasons 2018–2020) were recorded every minute by the meteorological observatory at Dumont d'Urville station (operated by Météo France) and the meteorological observatory at Neumayer station.

The meteorological data for Coulman Island, Atka Bay (season 2011), and Stancomb-Wills Glacier stem from the ECMWF model "The ERA5 global reanalysis", which provides 1 h temporal and 1 km² spatial resolution[33] and is available online (https://www.ecmwf.int/en/forecasts/dataset/ecmwf-reanalysis-v5).

## Phenological model

We developed a mechanistic phenological model to describe the temporary fluctuations of the number of penguins at the colony over the course of a breeding season. The model parametrizes the annual abundance curve (number of animals present per day) with 14 parameters (Table 1). With a set of values for those parameters, the number of male breeders, female breeders, non-breeding adults, and chicks can be calculated deterministically for every given time point in the breeding season.

We estimate the parameter values for each colony and season by fitting the model prediction of total adult animals (breeders and non-breeders) to the respective ground-truth count obtained by human analysts. The model parameters can be estimated by fitting either of the provided functions (male breeders, female breeders, non-breeders, chicks) or combinations of those. We use Markov chain Monte Carlo (MCMC, see Supplementary Note 1) sampling[43–45] to fit the model's parameters. The sampling process provides us with a distribution of parameter estimates, from which we can derive a mean (best estimate) and standard deviation (error). The best parameter estimates for each colony and each season are listed in Supplementary Table 1.

The model parameters define the mean (time point) and width (duration) of Gaussian-shaped distributions of phenological events (e.g. arrival, female departure, hatching) as well as the number of animals that enter or leave during the event. The number and order of events is fixed in the model structure. The cumulative distribution of an event multiplied by the number of animals (specific for male breeders, female breeders, non-breeders, and chicks) yields the number of individuals that have undergone the event and have entered (e.g. breeders after arrival) or left (e.g. females after laying) the colony. By summing over all cumulative distributions (accounting for departing events with a negative sign), we arrive at the number of present individuals (see Fig. 7).

We chose the number and type of phenological events observed in the model according to what is known about the breeding cycle and which events could be observed directly in the number of individuals (e.g. female departure is observable, egg-laying is not). The model contains the following phenological events:

- arrival of the animals (breeders and non-breeders) at the colony site at the beginning of the breeding season,
- departure of the females after courtship and laying an egg, and departure of the non-breeders after unsuccessful courtship,
- return of the females for chick feeding and departure of the males for foraging at sea after hatching of the chicks,
- subsequent (2 x, according to ref. 19) parental switching between feeding and foraging, departure of both parents after their chick becomes thermally independent,
- subsequent (7 x, according to ref.19) feeding and foraging trips (females and males) until fledging of the chicks.

Note, that the number of foraging trips (2 x during guarding phase and 7 x during crèching phase) represent the average over the whole colony. If the number of trips (and therefore the number of events) is chosen as a free parameter of the model, the fit does not converge anymore. We believe that this numeric instability is due to different scenarios (few long trips or many short trips) creating a multitude of local optima. We furthermore believe that the average number is

sufficient to describe the entire colony due to the large number of breeding pairs.

The number of individuals entering/leaving during each event depends on the category of individual (female breeder, male breeder, non-breeder, chick) and the following model parameters number of breeding pairs, ratio of hatching chicks to number of breeders, ratio of fledging chicks to number of breeding pairs, and ratio of non-breeders to breeding pairs. At arrival, all female breeders, male breeders and non breeders enter the colony, but after courtship only females and non-breeders leave. After the female return, all males leave, but also all females without a hatched chick. The parental switching during brooding is synchronized so that parents overlap for a short period of time. The chicks "enter" the colony and therefore add to the colony size at the time of their thermal emancipation (when it is possible to count them). During the chick rearing period, chick mortality is modeled as chicks and parents leaving the colony without returning after a foraging trip. Details on how the factors, event time and duration for each individual event depend on the model parameters are given in Supplementary Note 3. The time points of the two parental switching events (between feeding at the colony and foraging at sea) for each sex, and the 7 consecutive feeding and foraging trips, are not free parameters. Rather, they are determined by the duration of a foraging trip at sea and the duration of the stay at the colony for feeding the chick. Furthermore, we assume that both the foraging and feeding durations decrease over time, in line with field observations[17–22], which we model by two linear functions (see Fig. 8). Furthermore, the duration of the first foraging trip of the females is assumed to be shorter than the first foraging trip of the males, also in line with field observations[19,35].

Similarly, the standard deviations of all events are not free parameters but are constrained as follows: the standard deviation of arrival of all adult penguins, and the standard deviation of the first departure of females and non-breeders are the same, assuming that courtship is equally long for all breeders. The standard deviation of the first return of females and of all subsequent events are also equal and must be greater or equal than the standard deviation of arrival.

The parameter limits for the fit were not chosen in strict limitation to the observed phenological variance at Pointe Géologie (or Atka Bay), but chosen to have the widest possible range within physiological boundaries of the species. For example, the model does not assume the incubation period to have a physiologically fixed length on a population scale, but the absence of female breeders can not last >100 days.

The individual counts used for fitting were conducted at the Pointe Géologie (PG) emperor penguin colony in vicinity to the Dumont d'Urville French research station (66° 40'S, 140° 01'E) and at the Atka Bay (AB) colony in the vicinity to the Neumayer III German research station (70° 40'S, 8° 16'W). The data from Pointe Géologie were collected over 10 years (2012 to 2021), Atka Bay data were collected over 3 years (2018–2020) (Fig. 2). Weekly counts were acquired by imaging the colonies with either hand-held (Pointe Géologie) or remote-controlled cameras (Atka Bay[14],) from an elevated position, resulting in one or several panoramic images per day that have sufficient resolution to count individual penguins. For manual counting, we used Adobe Photoshop or Clickpoints[31].

For model benchmarking, manual observations of phenological events such as time of arrival, female departure, or the beginning of fledging were recorded only at Pointe Géologie. However, some events (e.g. beginning of fledging, see Supplementary Table 2) could not be recorded in some years for various reasons such as inaccessibility of the colony, visual obstruction of the colony by geographic landmarks, or a low number of breeding pairs.

The ground-truth numbers of breeding pairs at AB and PG were taken from individual counts during the incubation period when only the incubating males are at the colony. Lost eggs/chicks were counted daily at PG. The number of fledged chicks was not counted. Therefore, we used the last known number of chicks

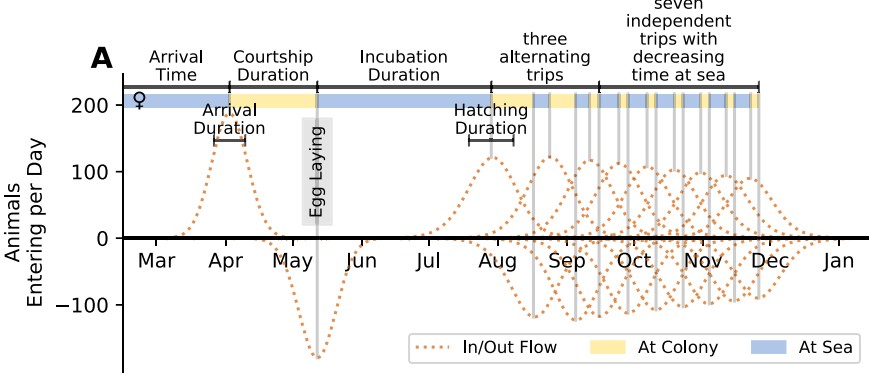

**Phenological Events**

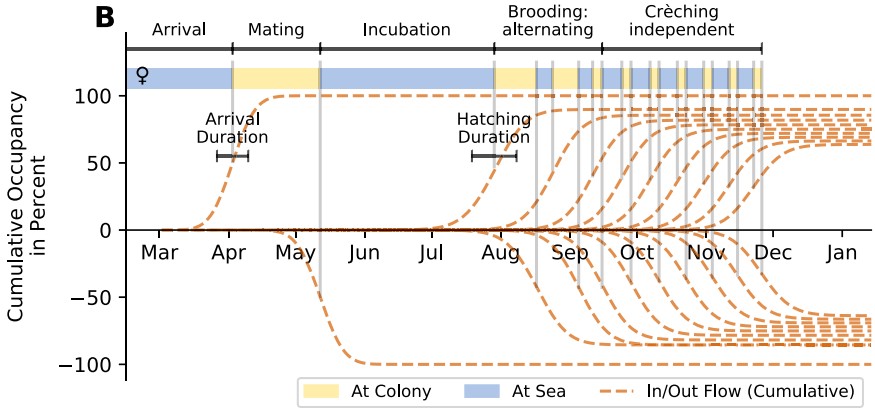

**Cumulative Flow of Individuals**

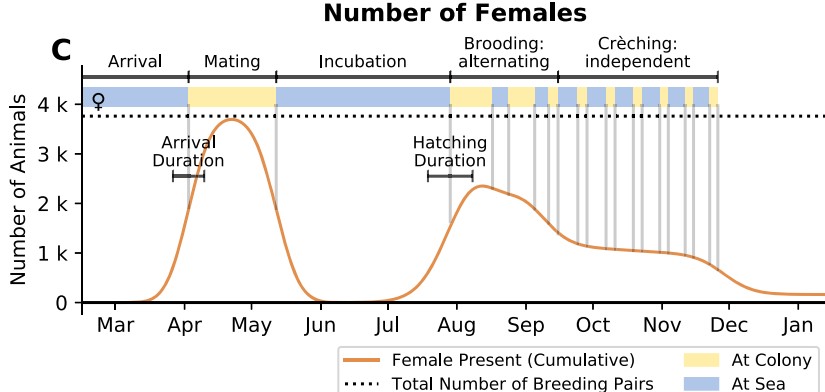

**Number of Females**

**Fig. 7 | Illustration of the breeding cycle and the phenological model (based on data from 2012 at Pointe Géologie).** The model describes the number of individuals present at the colony as a result of phenological events that bring individuals to return or leave the colony site. This pattern of presence and absence is represented by colored bars (yellow: at the colony, blue: at sea) in each of the plots. The model assumes that the probability densities for individual animals of returning and leaving the colony are normally distributed. The mean and width of each of these distributions are model parameters. **A** Probability density distribution of all phenological events included in the model. Arrivals are indicated by positive values, departures by negative values. The width of the distributions indicates the distributions of individual arrival or departure times. The height corresponds to the number of individuals participating in the event. **B** Cumulative probabilities (computed by integrating the probability density distributions) indicate the number of individuals over time per event as a percentage of the total number of breeding pairs. Positive values indicate arrivals, negative values indicate departures. Note that the number of adults decreases over time due to loss of eggs or chicks. **C** Projected number of females present at the colony, computed as the sum of the cumulative probabilities in (**B**) multiplied by the total number of breeding pairs (dotted black line). Source data are provided as a Source Data file.

before the onset of fledging as an estimate of the number of surviving chicks.

**Applying the phenological model to data from satellite images**
When we fit the phenological model to count estimates from satellite images, problems may arise due to the small number of usable images for each season (typically <10 images, available only during the months from September to December). The fit of 14 free model parameters to such a small number of data points can lead to numerical instability and large confidence intervals. Therefore, we fix all except three model parameters to the average values that we estimated from the model fit to ground-truth counts at Atka Bay and Pointe Géologie (see

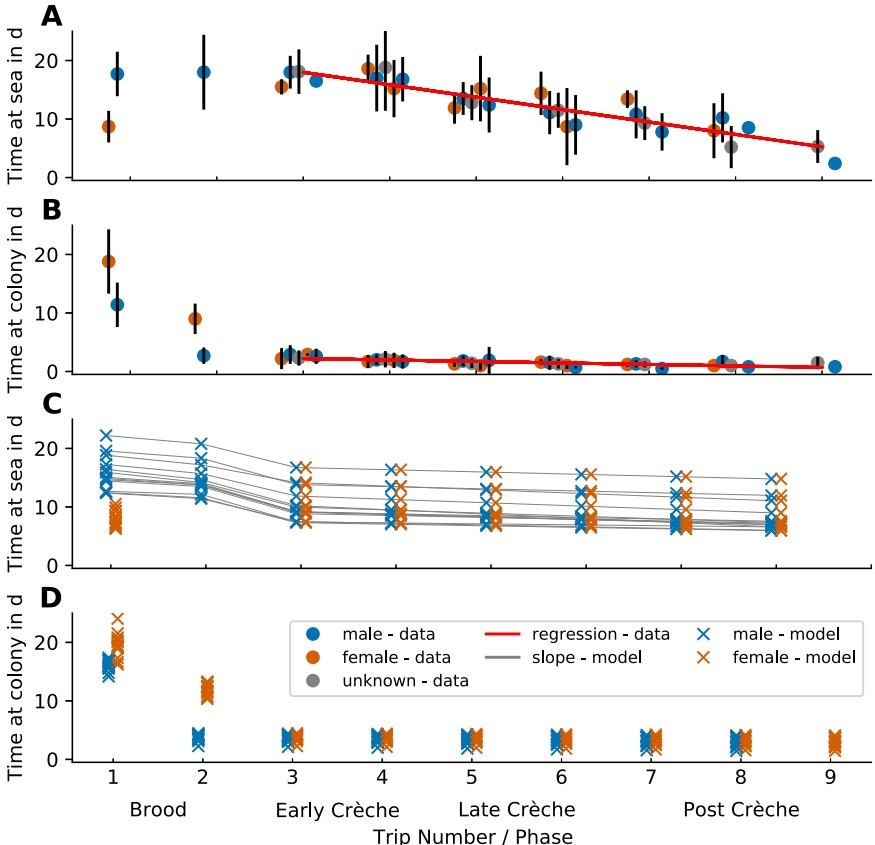

**Fig. 8 | Individual trip durations during chick rearing phase.** Time (in days, mean ± 1 standard deviation) a breeding emperor penguin spends at sea (**A**) or at the colony (**B**) at different stages of the breeding cycle (data from[19]). Red regression lines (**A**, **B**) show the decreasing trend in both time at sea and at the colony, after the initial brooding phase. Measurements were conducted at two colonies (Auster and Taylor Glacier). Standard deviations and mean values are derived from 9 (Auster, Female), 22 (Auster, Male), 26 (Auster, unspecified sex), 3 (Taylor, Female), and 8 (Taylor, Male) animals. **C**, **D** show the predicted time at sea and time at colony as predicted by the phenological model for 10 seasons (2012–2021) for Pointe Géologie and 3 seasons (2018–2020) for Atka Bay. Red, blue and gray dots denote the sex of the observed individuals (female/male/unknown). Gray lines indicate corresponding predictions for each season. Note that the model predicts large seasonal variations for the time at sea, but not for the time at colony. Source data are provided as a Source Data file.

Supplementary Table 1, Supplementary Fig. 7). The time of arrival is chosen so that the date of female return aligns with the first sunrise after mid winter +27.4 days. The only two free fit parameters are the number of breeding pairs and fledging success. This approach is justified by the fact that the phenology of the two colonies we have studied here (AB and PG) are similar except for the arrival time, as seen by the similar values of the model parameters and the absence of large systematic fluctuations during different years.

### Reporting summary
Further information on research design is available in the Nature Portfolio Reporting Summary linked to this article.

## Data availability
The individual count data, area measurements from images and satellites, the meteorological measurements, the results of the bayesian sampling, the manually observed phenological event dates, and the manually observed breeding success measures are available at GitHub (https://github.com/whoi-mars/EmperorPenguinPhenology)[46] and provided in the Supplementary Information/Source data file. Source data are provided within this paper. Source data are provided with this paper.

## Code availability
All software necessary to reproduce the study is publicly available in the repository https://github.com/whoi-mars/EmperorPenguinPhenology.

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

## Acknowledgements

This work was supported by the Deutsche Forschungsgemeinschaft (DFG) in the framework of the priority programme 1158 "Antarctic Research with comparative investigations in Arctic ice areas" by grants (FA336/5-1, ZI1525/3-1, ZI1527/7-1), by the Institut Polaire Français Paul-Emile Victor (IPEV) within the framework of the Project 137-ANTAVIA, by the Alfred-Wegener-Institut Helmholtz-Zentrum für Polar-und Meeresforschung (AWI) within the framework of the Projects SPOT (AWI_ANT_13) and MARE (AWI_ANT_14)[47], by the Centre Scientifique de Monaco with additional support from the LIA-647 and RTPI-NUTRESS (CSM/CNRS-UNISTRA), by the Centre National de la Recherche Scientifique (CNRS) through the Programme Zone Atelier Antarctique et Terres Australes (ZATA). We thank the former PIs of the IPEV project 137 (Y. Le Maho, S. Blanc). We also thank Météo France for the meteorological data of Dumont d'Urville. We are deeply grateful to all the wintering and summering members of projects IPEV 137, AWI-SPOT, AWI-MARE, and we also sincerely thank the IPEV and AWI logistics teams for their important and continued support in the field. This study was funded by the following programs: German Research Foundation grant FA336/5-1 (BF). German Research Foundation grant ZI1525/3-1 (DZ). German Research Foundation grant ZI1527/7-1 (DZ). National Science Foundation grant 2046437 (DZ). Woods Hole Oceanographic Institution (DZ). Institut Polaire Français

Paul-Emile Victor (IPEV) 137-ANTAVIA (CLB). Centre Scientifique de Monaco (CLB). Centre National de la Recherche Scientifique (CLB). CSM/CNRS-UNISTRA (LIA-647 and RTPI-NUTRESS) (CLB). The authors gratefully acknowledge the scientific support and HPC resources provided by the Erlangen National High Performance Computing Center (NHR@FAU) of the Friedrich-Alexander-Universität Erlangen-Nürnberg (FAU). The hardware is funded by the German Research Foundation (DFG).

## Author contributions

Conceptualization: A.W., S.R., B.F., C.L.B., D.Z. Methodology: A.W., S.R., B.F., A.M., C.M., C.L.B., D.Z. Data analysis: A.W. Data collection: A.W., S.R., A.H., T.B., M.B., C.C., D.C., R.C., C.E., B.F., A.K., A.M., D.M., J.M., S.P., E.S., C.L.B., D.Z. Supervision: B.F., C.L.B., D.Z. Writing—original draft: A.W., S.R., A.H., B.F., C.L.B., D.Z. All authors contributed critically to the drafts and gave final approval for publication.

## Funding

## Competing interests

The authors declare no competing interests.

## Ethics

Health, safety, security and other risks for participating researchers were assessed and managed by the institutions that provided logistic support at the research stations involved: Institut Polaire Français Paul-Emile Victor (IPEV) for Dumont d'Urville (Pointe Géologie) and Alfred-Wegener-Institut Helmholtz-Zentrum für Polar-und Meeresforschung (AWI) for Neumayer III (Atka Bay). Ethics questions concerning local and regional researchers, partners, or governments do not apply, due to Antarctica being uninhabited.
