## [Peer Review File · Nature Communications]

Reviewers' Comments:

Reviewer #1:

Remarks to the Author:

This is a huge body of work that is very relevant to the conservation of emperor penguins. Especially the fact that the mechanistic models the authors have built can estimate number of chicks with only information about adult counts and breeding phenology of the species is important. I have, however, several concerns, criticisms and suggestions.

1 – There are a lot of moving parts in the manuscript, especially methods wise, and their connection can sometimes be hard to follow. I believe this can be improved by re-ordering results sections and following this same order in the methods. My suggested order is below but this can potentially be improved further.

There are 3 main methodological parts of the manuscript, the phenological model, the windchill model, and area to colony area calculation. The authors start by describing the phenological model and its results and spend quite a few pages on it before moving to the remote sensing components. This can be confusing to the reader because the expectation is to read a paper about remote sensing and this is clearly stated as the main goal of the paper. So, reading the phenological part first makes it feel like this is a completely different paper than advertised. The link between the phenology and remote sensing becomes only clear at the very end of the results section.

My suggestion is simple, first start by describing colony area detection, then move to the windchill model by making the connection that density of the colony depends on local meteorological conditions and this density affects colony size estimates by remote sensing. You can end this section by hinting at the readers that such a windchill model can also be used by satellite imagery. After this you can proceed to the phenological model, again reminding the reader that such a model is necessary because chicks and breeding success cannot be remotely determined. Explicitly mention that you develop this model first using ground counts then apply it to the satellite imagery. Finally, at the last section you can describe how the windchill model, area to colony size conversion and the phenological model comes together with satellite imagery.

I believe that you can move the results about the phenological parameters to supplementary information because it breaks the narrative. You should also move the section about predictive performance of the phenological model to the main phenological model section. So only three sections in the results section are enough in my opinion: Colony size estimation with the windchill model, phenological model, application to satellite imagery. Finally, please rename titles of the results sections to be more informative to a general ecological audience. For example, "Converting colony area to colony size (windchill model)", "Estimating breeding success from adult penguin counts (phenological Model)", "Applying the windchill and phenological models to satellite imagery".

2 – My main concern is that the presentation of the phenological model is very obscure. The verbal description provided in the main text is generally clear but certain details are hard to follow (more on this below). A full mathematical description of the model is necessary. The "math" presented in the supplementary information reads more like someone's comments to their own code rather than a full mathematical model. I suspect, in its current state, it will be a black box for a typical audience of this journal.

While Fig. 7 does a good job of showing the basic structure of the mechanistic model, it is very hard for me to understand how this model was parameterized. Table 1 in this sense is not helpful, I'm not sure authors intended it to be on this state but right now it looks haphazardly made because it stretches across multiple pages. Also, are all the parameters estimated on Table 1? Do you only provide upper and lower limits to the model? In Fig. 7, we see that durations and timings of breeding events have a gaussian distribution with an associated mean and standard deviation. Are these parameters estimated, or fed to the model? The model fits each year separately, so where is the variability of arrival time and all other parameters coming from? Is that variability assumed to be the annual variability observed by the authors during their data collection on the

field? Finally, how exactly are the linear relationships in Fig. 8 used in the phenological model?

I can just continue writing questions, but overall, the way phenological model section in methods is structured makes it really hard to understand the model itself because the information provided does not follow a clear narrative. It reads more like a randomly distributed facts about the model. Paragraph on line 505 is a very good example:

“The model parameters (time point, standard deviation, and amplitude of each event) are determined by a least squares fit of the model predictions to data. Since observers usually cannot distinguish between male and female animals, the sum of all adult animals is considered. For fitting the model, the number of chicks is not considered, however if this number is available from observations, it can be used for benchmarking the predictive power of the model.”

I don't know what it means to determine “time point, standard deviation, and amplitude of each event by a least squares fit of the model predictions to data”. The second sentence is about using sum of all adult animals and the last sentences is about using chicks for benchmarking predictions. It seems to me that all these three sentences in this paragraph is talking about a completely different aspect of the modelling process. This is the general state of the methods section. I strongly recommend a complete rewrite of the whole phenological model section where each information provided to the readers is built on the information previously provided following a clear narrative. Otherwise it is impossible to judge the validity of this mechanistic model.

3 – Authors report generally favorable results for their phenological model but it is not immediately clear to me how useful this model is beyond the application presented here.

This is a highly parametrized model with a lot of assumptions where each parameter has clear limits (upper and lower bounds). The information for the majority of these assumptions and parameter limits are coming Point Geologie (not just this manuscript but the broader emperor penguin literature as well). So, is it really surprising that adult counts and model estimation of colony size fit so well? The model could be so overfit that practically what is being done could be just connecting the dots across adult counts. So, I believe that the generalizability of this model to other colonies is questionable. Each year is fit independently, so a temporal would be unnecessary but a spatial one is feasible, where a model fit on the 2018 data of Point Geologie is tested on counts from Atka Bay. This can be repeated with years 2019 and 2020.

Authors perform a similar procedure with data from other colonies namely, Coulman Island and Stancomb-Wills. The model's predictions for these colonies are not all bad but it is not completely promising either, predicting nearly double the amount for Stancomb-Wills and more than 5000 individuals less for Coulman Island (Fig. 6G). The fact that its predictions for Point Geologie and Atka Bay are much better reinforces my suspicion that there are limits to the generalizability of the phenological and the windchill models. Finally, unlike the phenological model, there is no information regarding the generalizability of the windchill model, which is transferred across both space and time.

4 – Figs. 3A and 3B are hard to read. I don't know what difference between “Breeding Pair” and “True is”, as they have the same symbol. What is being predicted here, breeding success or breeding pairs? The text reads breeding success but y axis is number of individuals. The predictions are hard to see also among so many colors. I recommend a different figure format to convey this information.

Fig. 3 is important for providing information about how well the phenological model fits the general phenology of the species, but it has important issues. First, the strong correlation of the larger figure in Fig 3C is a self-fulfilling prophecy. The model has all the information regarding the potential start and end of every event. Even if these parameters have upper and lower bounds the model knows that arrival cannot happen in January. So, this strong fit on the larger figure does not provide any information about model performance, it's simply the natural result of the information provided to the model. Also, I don't find the smaller figures as successful as authors make it to be. Depending on when an event happens in a given year changes how and in what direction the model is going to be biased.

Reviewer #2:

Remarks to the Author:

Review of Remote sensing of emperor penguin abundance and breeding success

This paper is an extremely useful method for refining the estimation of emperor penguins from optical satellite imagery methods currently used by several groups in Antarctic science.

It is an important step forward in understanding their populations and future population trends.

The developed methods are the result of several years of fieldwork and study towards an important goal. Well done to those involved. Generally the paper is well written, the methods and flow understandable and the figures clear. The methods are generally good, although further report of the potential errors in the method would be useful for fuller understanding of the results. There are a few cases where the assumptions are leaned on a little too heavily and I would suggest that in a number of cases where the authors use the term "probably", I would replace it with "possibly". The main concern with the method is that it is from only two sites, and while these represents different areas of the continent, they certainly do not replicate all the geographical variability amongst emperor penguin colony locations. Although this is not a criticism of the method, it is an important point and needs to be considered in the discussion. It is especially relevant to the phenology results as we know from published and unpublished data in the Ross Sea, that phenological events can be different – it seems that fledging may happen earlier in the Ross Sea where we have some data from Cape Crozier. Likewise, we have absolutely no data from the West Antarctic (except very early reports from Stonehouse on the Dion Islands which is on the Peninsula) and as these colonies are already experiencing sea-ice loss it is unknown whether phenological timings have adapted in any way to the changes. I would suggest a short paragraph in the discussion section noting this and also suggesting further work in other areas.

I have listed below a number of smaller comments and revisions.

Line

43. the species is predicted to become extinct by the end of this century. – this is incorrect. Latest predictions from Jenouvrier, suggest that at present rates of greenhouse gas emissions (the high end 8.4 IPCC scenario) we will lose ~90% of colonies by the end of the century.

43. A significant decline of sea ice could already be observed. I would change this to "has been observed"

74 is on the order of 0.5 to 1 m/pixel – current maximum resolution is 0.3 m for optical and 0.15 m for SAR imagery. I would change the following sentence from "cannot yet" to "have not yet" provided measurements at an individual level.

136 should the 20% term in the sentence; an overestimate of 25% (multiply by 1.25) or an underestimate of 20% (divide by 1.25). be 25%? Or am I missing something here? If so a fuller explanation is needed.

140 Figure 1 – I suggest expanding the grey fledging bar to the last week of September, as, at Atka Bay, it seems that the decline in chick numbers starts around 24th December rather than 1st of January as the grey bar suggests.

171 eggs with a mean absolute error of +/- 190 animals. – It would be useful to note this as a percentage as well.

Fig 3 – it seems that the dates on fig3c do not match with the grey bars on figure 1 -please check these. It would be useful to have dates on the gridlines of the inset boxes as they are very difficult to interpret otherwise.

216 where does the 12.8 figure come from and how variable is it?

278 "The model accurately predicts the number of breeding pairs and number of fledging chicks for both colonies". This sentence is repetitive and should be removed.

280 the statement "probably due to a lower number of sample counts 281 for Atka Bay than for

Pointe Géologie (26 vs 44 counts per season).” Is an assumption based on very little data. With only two sample points, AB and PG, it is impossible to say if this is the case. It would be useful to note this in the text, maybe saying that the lower values at Atka bay could be due to the low sample size, but they could also be due to unknown variance in phenology at the two sites. Further studies at other colonies would confirm these assumptions.

355 Insert the word optical here: “Optical satellite images, currently used to estimate populations, can only be taken during periods when occupancy is most erratic”

392-395 I would be a little more cautious with the statement “This increase by 393 more than 2,000 breeding pairs is likely due to an immigration of individuals from the 394 neighboring Halley colony that experienced three consecutive years of breeding failure (from 395 2014 to 2016) and had completely abandoned the colony site in 2016.” I would agree that the displacement of adults from Halley could be the root cause of the increase at Atka, but it is possible that the increase was juvenile recruitment from Halley or displacement of adults from other colonies closer to Halley, where Halley birds had settled. Overall I think it would be more correct to say that the increase in numbers could have been due to immigration, recruitment or displacement, caused by the almost complete abandonment of the Halley Bay colony between 2016 and 2018.

435 On the perspective correction method, do the authors have any statistics on the accuracy of this method compared to drone analysis? It would be useful to get an idea of what the potential error is in the method.

465 On the statement: “The most important model parameter is the number breeding males and females. This number is assumed to remain constant during a breeding cycle...” Is there any consideration or analysis of the percentage of non-breeders at the colony? Are all birds breeders? Also, when egg or chick mortality happens surely many breeders leave? At PG there can be over 30% chick mortality (old reports from Prevost), so does this number of breeders really remain constant? I think that this fact is mentioned later in the methods around line 490, but that goes against the original statement, so you may need to change or expand the statement.

REVIEWER COMMENTS

Reviewer #1 (Remarks to the Author):

This is a huge body of work that is very relevant to the conservation of emperor penguins. Especially the fact that the mechanistic models the authors have built can estimate number of chicks with only information about adult counts and breeding phenology of the species is important. I have, however, several concerns, criticisms and suggestions.

1

– There are a lot of moving parts in the manuscript, especially methods wise, and their connection can sometimes be hard to follow. I believe this can be improved by re-ordering results sections and following this same order in the methods. My suggested order is below but this can potentially be improved further.

There are 3 main methodological parts of the manuscript, the phenological model, the windchill model, and area to colony area calculation. The authors start by describing the phenological model and its results and spend quite a few pages on it before moving to the remote sensing components. This can be confusing to the reader because the expectation is to read a paper about remote sensing and this is clearly stated as the main goal of the paper. So, reading the phenological part first makes it feel like this is a completely different paper than advertised. The link between the phenology and remote sensing becomes only clear at the very end of the results section.

My suggestion is simple, first start by describing colony area detection, then move to the windchill model by making the connection that density of the colony depends on local meteorological conditions and this density affects colony size estimates by remote sensing. You can end this section by hinting at the readers that such a windchill model can also be used by satellite imagery. After this you can proceed to the phenological model, again reminding the reader that such a model is necessary because chicks and breeding success cannot be remotely determined. Explicitly mention that you develop this model first using ground counts then apply it to the satellite imagery. Finally, at the last section you can describe how the windchill model, area to colony size conversion and the phenological model comes together with satellite imagery.

Following the reviewer's suggestions, in the Results section we combined all details about the phenological model into one coherent block, leaving only 3 subsections: 1 Windchill model, 2. Phenological Model, 3. Application to satellite imagery. We added a short paragraph at the end of the introduction explaining the structure of the results section:

“The method can be broken down into three separate steps. First, we convert colony covered areas from ground based or satellite imagery to individual counts by modeling the

colony density as a function of temperature, wind speed, solar radiation, and humidity at the colony site (“windchill model”). Second, we present a phenological model that describes how the number of individuals present at the colony on each day depends on the number of breeders and the breeding success. We benchmark the model with ground based individual counts. Third, we invert the phenological model to infer the number of breeding pairs and the breeding success from sparse counts of adult animals at the site of the colony, obtained near the end of the breeding season.. We benchmark this method with data from ground-based and satellite-based images.”

We further moved the paragraph describing the phenological model’s performance in predicting the number of breeding pairs and the breeding success closer to the beginning of the paragraph to highlight these key findings.

I believe that you can move the results about the phenological parameters to supplementary information because it breaks the narrative.

We understand the reviewer's concern about the narrative of the manuscript. The list of "phenological parameters" (Table 1) could indeed be moved to the SI, but we feel that having this list of parameters available without first having to search for it in the SI, which we know is a barrier for some readers, may actually make it easier to follow and understand the phenological model. But we have no strong feelings, and if the editor sides with the reviewer, we can move Table 1 to SI.

You should also move the section about predictive performance of the phenological model to the main phenological model section. So only three sections in the results section are enough in my opinion: Colony size estimation with the windchill model, phenological model, application to satellite imagery.

We agree and have changed the structure of the results paragraph accordingly as described above.

Finally, please rename titles of the results sections to be more informative to a general ecological audience. For example, “Converting colony area to colony size (windchill model)”, “Estimating breeding success from adult penguin counts (phenological Model)”, “Applying the windchill and phenological models to satellite imagery”.

We renamed the titles of the results section according to the reviewers suggestion.

2

– My main concern is that the presentation of the phenological model is very obscure. The verbal description provided in the main text is generally clear but certain details are hard to follow (more on this below). A full mathematical description of the model is necessary. The “math” presented in the supplementary information reads more like someone’s comments to their own code rather than a full mathematical model. I suspect, in its current state, it will be a black box for a typical audience of this journal.

We agree and provide more details of the mathematical model along with a hopefully more understandable description of the model.

While Fig. 7 does a good job of showing the basic structure of the mechanistic model, it is very hard for me to understand how this model was parameterized. Table 1 in this sense is not helpful, I'm not sure authors intended it to be on this state but right now it looks haphazardly made because it stretches across multiple pages.

We changed the format of Table 1 to fit it into one page.

Also, are all the parameters estimated on Table 1?

Yes, all parameters listed in Table 1 can be estimated from the fit of the model to the manual counts, which we have done for example in Fig. 3. However, when we applied the model to the satellite based counts, the small number of measured data points and their large variability make it difficult to robustly fit all 14 free parameters at once. Therefore, only BP (number of breeding pairs) and F (fledging success ratio) were fitted to the data. t_0 (arrival date) was inferred from our observation of the latitudinal shift. All other parameters were fixed to their mean values as estimated from the manual counts.

Do you only provide upper and lower limits to the model?

Yes, the model is only provided with upper and lower bounds for the parameters, and a starting value chosen between the upper and lower bounds. The upper and lower bounds for the parameters are chosen with a wide margin around the values observed at Pointe Géologie and Atka Bay, so that the model can also capture the phenology of other colonies.

In Fig. 7, we see that durations and timings of breeding events have a gaussian distribution with an associated mean and standard deviation. Are these parameters estimated, or fed to the model?

The distributions and timing of phenological events are estimated by the model, but only when we fit the model to the manual counts (Fig. 3). When we apply the model to satellite-based counts, the small number of measured data points and their large variability make it difficult to robustly fit all 14 free parameters at once. Therefore, only BP (number of breeding pairs) and F (fledging success ratio) are fitted to the data. t_0 (arrival data) is inferred from our observation of the latitudinal shift. All other distribution and timing parameters are fixed to their mean values as estimated from the manual counts, i.e. they are fed to the model.

The model fits each year separately, so where is the variability of arrival time and all other parameters coming from?

The parameter variance is estimated from the variance-covariance matrix, which in essence describes the parameter uncertainty, or put differently, how much the quality of the model fit to the data for a given year is influenced by small changes to each of the model parameters. We numerically estimate the variance-covariance matrix using a Markov-Chain Monte-Carlo sampling method (*Salvatier et al 2016, Hoffmann & Gelman 2014*), which provides 2200 samples for every fit parameter for every year. The reported variability is the standard deviation of those 2200 samples.

Is that variability assumed to be the annual variability observed by the authors during their data collection on the field?

Apart from the parameter uncertainty explained in the previous paragraph, we assume that the individual counts of each day are erroneous, with a log-normal distributed error around the true number of individuals on that day. We do not set the width of this distribution, but estimate it

during the model fitting process. This process provides us with an estimate of the counting error, when counting emperor penguins from panoramic images .

Finally, how exactly are the linear relationships in Fig. 8 used in the phenological model?

The model does not include individual parameters for each foraging trip/ feeding duration, as this would increase the number of free parameters and thus increase their uncertainty. The model specifies only the first foraging and feeding duration (s_{\max} and c_{\max}) and the last foraging and feeding duration (s_{\min} , c_{\min}). These values define the linear relationship between trip number and duration. The equations are given in S9.

I can just continue writing questions, but overall, the way phonological model section in methods is structured makes it really hard to understand the model itself because the information provided does not follow a clear narrative. It reads more like a randomly distributed facts about the model. Paragraph on line 505 is a very good example:

“The model parameters (time point, standard deviation, and amplitude of each event) are determined by a least squares fit of the model predictions to data. Since observers usually cannot distinguish between male and female animals, the sum of all adult animals is considered. For fitting the model, the number of chicks is not considered, however if this number is available from observations, it can be used for benchmarking the predictive power of the model.”

This is indeed a good example of bad writing. We apologize. We have rewritten much of the text to improve sentence and paragraph structure.

I don't know what it means to determine “time point, standard deviation, and amplitude of each event by a least squares fit of the model predictions to data”. The second sentence is about using sum of all adult animals and the last sentences is about using chicks for benchmarking predictions. It seems to me that all these three sentences in this paragraph is talking about a completely different aspect of the modelling process.

"Least squares fit" is the technical term for varying the model parameters until the sum of the squared differences between the model predictions and the data is minimized. See the Methods section for more details.

This is the general state of the methods section. I strongly recommend a complete rewrite of the whole phenological model section where each information provided to the readers is built on the information previously provided following a clear narrative. Otherwise it is impossible to judge the validity of this mechanistic model.

We agree and have re-written large parts of the text to improve sentence and paragraph structure. Accordingly, we rewrote the phenological model section in the methods section (lines 494 to 554 in the revised manuscript).

3

– Authors report generally favorable results for their phenological model but it is not immediately clear to me how useful this model is beyond the application presented here.

This is a highly parametrized model with a lot of assumptions where each parameter has clear limits (upper and lower bounds). The information for the majority of these assumptions and parameter limits are coming Point Géologie (not just this manuscript but the broader emperor penguin literature as well).

We fully agree that this is a valid concern. Literature knowledge about emperor penguins's breeding phenology is limited, as it is mainly based on Pointe Géologie. Regarding assumptions about parameter limits: We did not set the parameter limits based to the observed inter-year variation at Pointe Géologie (or Atka Bay), rather, we chose parameter limits that were as non-informative as possible and as large as reasonable without grossly violating animal physiology. For example, a range (around the mean) of 100 days for arrival, a range of 21 days for courtship, a range of 50 days for female absence (incubation), a range of 5 days of stay at colony for feeding (not including time spent waiting for partner during guarding), up to 21 days at sea. In addition, those time ranges are cumulative: Arrival can happen between Feb 20 and May 31 (range 100d), Female departure between Mar 20 and July 12 (range 114d), female return between May 09 and Oct 20 (range 164d), etc. To better explain our reasoning, we added the following sentence to the phenology model section in Methods:

“The parameter limits for the fit were not chosen in strict limitation to the observed phenological variance at Pointe Géologie (or Atka Bay), but chosen to have the widest possible range within physiological boundaries of the species. For example, the model does not assume the incubation period to have a physiologically fixed length on a population scale, but the absence of female breeders can not last longer than 100 days.”

So, is it really surprising that adult counts and model estimation of colony size fit so well? The model could be so overfit that practically what is being done could be just connecting the dots across adult counts.

Avoiding overfitting was one of our main concerns. Our arguments why our model does not overfit the data despite having 14 free parameters are as follows:

First, the goal of our model - and indeed of most models - is not to connect the dots, but to describe the "true" data without describing the inherent noise. The structure of our model is such that it can only generate a smooth large bump-shoulder-small bump-tail curve (Fig. 3); it cannot describe other shapes and cannot account for the noisy "wiggles" in the data points. A model that overfits the data, by contrast, would try to account for the noise and be influenced by it.

Second, our model parameters not only have a clear physiological meaning, they also have a very specific role in describing the large bump-shoulder-small bump-tail phenology curve. For example, arrival time, its variance, and the total number of adults describe the first large bump in the phenology curve, and none of the other parameters can affect its shape. At the same time, by fitting the model to just the first few data points, we can already learn about 3 important phenological parameters - arrival time, its variance, and the total number of adults. If we had described the phenology curve with, say, a polynomial function, we might have been able to fit the data with fewer parameters, but we would not have learned much about phenology.

Third, the variance (uncertainty) of the fit parameters is low. This would not have been the case in a model that overfits the data.

Fourth, the model makes accurate predictions about the number of chicks and breeding success. These data were not used to fit the model. Based on this predictive power, we have done extensive benchmarking and can say with confidence that the model does not overfit the data.

Fifth, overfitting is only a concern in cases where we have few data points, specifically in the case of a few satellite images towards the end of the breeding season. In this case, we fit only two free parameters (number of breeding pairs, fledgling success ratio) and fix all other parameters.

So, I believe that the generalizability of this model to other colonies is questionable. Each year is fit independently, so a temporal would be unnecessary but a spatial one is feasible, where a model fit on the 2018 data of Point Geologie is tested on counts from Atka Bay. This can be repeated with years 2019 and 2020.

We tested this idea and applied the parameter values from PG to AB and vice versa. The results are shown in the supplement (S23). We find that the spatial cross-correlation is very low (all R^2 values lower than 0.05), which means that the model predictions are specific for each year and each colony. This test is not a proof of generalizability, but it shows that the different phenologies between colonies at PG and AB are captured by the model.

Authors perform a similar procedure with data from other colonies namely, Coulman Island and Stancomb-Wills. The model's predictions for these colonies are not all bad but it is not completely promising either, predicting nearly double the amount for Stancomb-Wills and more than 5000 individuals less for Coulman Island (Fig. 6G). The fact that its predictions for Point Geologie and Atka Bay are much better reinforces my suspicion that there are limits to the generalizability of the phenological and the windchill models.

These discrepancies are less a problem of the model than of the ground truth data. In general, we consider the ground truth numbers given for the satellite counts to be more of an order of magnitude indicator than a representation of the actual number of breeders, because they were recorded in different years, up to decades earlier (Stancomb-Wills 1986, Atka Bay 1987, Coulman 2006), with the stage of the phenological cycle of the colony at the date of counting either undocumented or unpublished. If documented at all, the studies were conducted between October and November, and therefore the observed values may be quite different from the actual number of breeders. We have added a Table (S22) with different literature values and study conditions to the SI, and we have added the following paragraph to the discussion:

“We find rather high discrepancies between our estimates and literature values for the satellite-based analysis (Atka Bay -700 animals, Stancomb Wills +3400 animals, Coulman Island -5832 animals). However, we consider the literature values to have a very large uncertainty because they stem from ground count or aerial photography studies conducted years or decades earlier, during austral summer, without correction for phenological variance and colony occupancy (23). See Table S22 for a comparison of different censuses.”

Finally, unlike the phenological model, there is no information regarding the generalizability of the windchill model, which is transferred across both space and time.

The windchill model as described in our manuscript and also in Richter et al. 2018, is a considerably simple tool to describe an Emperor penguin colony's response to weather. Realistically this response is driven by the individual's physiology, which we expect to be very similar across colonies, due to their genetic closeness (Cristofari et al 2016). In addition, the response to cold will depend on the thickness of the animals' insulating fat layer, which changes drastically throughout the course of a breeding season and is likely very different between the same day in different years. We added the following paragraph to the discussion:

“The huddling behavior of emperor penguins and hence the colony density not only depends on weather conditions but also on the thickness of the animals' insulating fat layer. Therefore, the parameters of the windchill model will change over the course of the breeding season as the males lose body fat during their incubation time, or as well-fed females return to the colony. Moreover, seasons with poor food availability will result in lower individual fat reserves, which would also be reflected in altered model parameters. Currently, we have not explored the relationship between fat layer thickness and changes in windchill model parameters. The single set of parameters available today that we used in our study can accurately predict colony density at the end of the breeding season for the Atka Bay and Pointe Géologie colonies, but may not be fully representative for other times in the breeding season, and possibly also not for different colonies.”

4

– Figs. 3A and 3B are hard to read. I don't know what difference between “Breeding Pair” and “True is”, as they have the same symbol. What is being predicted here, breeding success or breeding pairs? The text reads breeding success but y axis is number of individuals.

The figure shows the ground truth and predicted values for the number of individuals (lost eggs, dead chicks, fledging chicks, and breeding pairs). We removed the word “breeding success” from the text to avoid confusion.

The predictions are hard to see also among so many colors. I recommend a different figure format to convey this information.

We realize that the figure legend might be difficult to read. We made the number of breeding pairs an additional bar on each side (predicted/observed) of the red/orange/green bar. Therefore, we keep the visual comparability between the sum of lost eggs, dead chicks, and fledging chicks, while deconvoluting the information and reducing the inset legend complexity. We changed the text legend to the following:

“Observed and predicted number of breeding pairs (blue), lost eggs (orange), dead chicks (red), and fledging chicks (green) by year and colony. For each year, the two left bars with a solid line show the observed numbers, and the two right bars show the predicted numbers. The predicted numbers of lost eggs, dead chicks, and fledging chicks add up to the number of breeding pairs by model definition, while the ground-truth values do not, due to counting inaccuracies.”

Fig. 3 is important for providing information about how well the phenological model fits the general phenology of the species, but it has important issues. First, the strong correlation of the larger figure in Fig 3C is a self-fulfilling prophecy. So, this strong fit on the larger figure does not provide any

information about model performance, it's simply the natural result of the information provided to the model.

The solid line is not a fit (it is the line of identity), and we actually don't compute a correlation between the measured and predicted data points displayed in Fig. 4C. This would indeed be a self-fulfilling prophecy and completely meaningless, as the spread of the data over a time course of 1 year is much larger than the variance between the predictions and the data. Rather, information about how well the phenological model fits the general phenology of the species is presented in Fig. 3 and Fig. 4 A,B. Key information is actually presented in the insets, and the larger figure is mainly intended for orientation.

The model has all the information regarding the potential start and end of every event. Even if these parameters have upper and lower bounds the model knows that arrival cannot happen in January. The margins on the parameter limits were much larger than the variance in the observed and inferred dates. Arrival could not happen in January, but between 20th of February and 31th of May, which allows for a large range of options for the model to fit.

Also, I don't find the smaller figures as successful as authors make it to be. Depending on when an event happens in a given year changes how and in what direction the model is going to be biased. These insets highlight the (dis)agreement between the model estimates versus estimates from observers on the ground. We account the obvious discrepancies not to a flaw of the model but to the low temporal resolution of the data acquisition by found-based observers (1 count per week):

“We could not quantify how well the model can predict annual variations in phenological events at the same colony, because most events vary less than the resolution of ground-based observations (7 days, see Fig. 4C).”

Reviewer #2 (Remarks to the Author):

Review of Remote sensing of emperor penguin abundance and breeding success

This paper is an extremely useful method for refining the estimation of emperor penguins from optical satellite imagery methods currently used by several groups in Antarctic science.

It is an important step forward in understanding their populations and future population trends.

The developed methods are the result of several years of fieldwork and study towards an important goal. Well done to those involved. Generally the paper is well written, the methods and flow understandable and the figures clear. The methods are generally good, although further report of the potential errors in the method would be useful for fuller understanding of the results. There are a few cases where the assumptions are leaned on a little too heavily and I would suggest that in a number of cases where the authors use the term “probably”, I would replace it with “possibly”.

We followed the reviewer's suggestion and replaced “probably” with “possibly”.

The main concern with the method is that it is from only two sites, and while these represents different areas of the continent, they certainly do not replicate all the geographical variability amongst emperor penguin colony locations. Although this is not a criticism of the method, it is an

important point and needs to be considered in the discussion. It is especially relevant to the phenology results as we know from published and unpublished data in the Ross Sea, that phenological events can be different – it seems that fledging may happen earlier in the Ross Sea where we have some data from Cape Crozier. Likewise, we have absolutely no data from the West Antarctic (except very early reports from Stonehouse on the Dion Islands which is on the Peninsula) and as these colonies are already experiencing sea-ice loss it is unknown whether phenological timings have adapted in any way to the changes. I would suggest a short paragraph in the discussion section noting this and also suggesting further work in other areas.

We agree and have added the following paragraph to discussion:

“How well the model performs the timing of phenological events is difficult to assess because ground-based observations are only intermittent and moreover based on single events of individual animals, for example the first penguin arriving at the breeding site. By contrast, the phenological model takes all available counts into consideration and from that computes the central tendency of events, e.g. the peak arrival time. The temporal agreement between model predictions and observations are within the sampling interval of about one week for ground-based observations, plus the timing offset between the first occurrence and the event peak. We tested our model against a second colony location (Atka Bay) and found a similarly good agreement, despite the model being mainly built on behavioral knowledge from Pointe Géologie. Nonetheless, additional ground-truth data from other colonies will help confirm the generalizability of the model.”

I have listed below a number of smaller comments and revisions.

Line

43. the species is predicted to become extinct by the end of this century. – this is incorrect. Latest predictions from Jenouvrier, suggest that at present rates of greenhouse gas emissions (the high end 8.4 IPCC scenario) we will lose ~90% of colonies by the end of the century.

Thank you, we changed the statement in the manuscript:

“The extent of the Antarctic fast ice is predicted to rapidly decline in the coming decades (8), and the species is predicted to lose 90% of its colonies by the end of this century”

43. A significant decline of sea ice could already be observed. I would change this to “has been observed”

We changed the line accordingly.

74 is on the order of 0.5 to 1 m/pixel – current maximum resolution is 0.3 m for optical and 0.15 m for SAR imagery. I would change the following sentence from “cannot yet” to “have not yet” provided measurements at an individual level.

We changed the statement accordingly and changed the lower limit of 0.5 m to 0.3 m.

136 should the 20% term in the sentence; an overestimate of 25% (multiply by 1.25) or an underestimate of 20% (divide by 1.25). be 25%? Or am I missing something here? If so a fuller explanation is needed.

20% was correct, because 100 divided by 1.25 is 80, hence a 20% underestimation. To avoid this confusion that a -25% geometric error corresponds to a -20% arithmetic error, we have re-phrased the paragraph:

“ Because the count data usually vary on a logarithmic scale, we report not arithmetic, but geometric errors: for example, a $\pm 25\%$ geometric error corresponds to a 1.25-fold overestimate (multiply by 1.25) or a 1.25-fold underestimate (divide by 1.25).”

140 Figure 1 – I suggest expanding the grey fledging bar to the last week of September, as, at Atka Bay, it seems that the decline in chick numbers starts around 24th December rather than 1st of January as the grey bar suggests.

The reviewer's observation is correct. However, we purposefully did not plot the gray bars in panels C & D (Atka Bay counts) because the indicated event timings were recorded at Pointe Géologie. Therefore we keep the plot as is.

171 eggs with a mean absolute error of ± 190 animals. – It would be useful to note this as a percentage as well.

We thank the reviewer for spotting the inconsistency. We added the percentage of 28% to the text: “and average geometric errors of 28%”

Fig 3 – it seems that the dates on fig3c do not match with the grey bars on figure 1 -please check these. It would be useful to have dates on the gridlines of the inset boxes as they are very difficult to interpret otherwise.

The apparent mismatch between Fig. 1 A&B and Fig 3 C may be due to different time scale ticks (Apr,Jun,Aug) vs. (Mar, May, Jul.). We added labels to the inset grid in Fig 3C.

216 where does the 12.8 figure come from and how variable is it?

This is a theoretical assumption. The densest possible packing of 30 cm diameter circles is 12.8 1/m^2 . Of course the chosen diameter does not represent all penguins in all stages of the phenological cycle well, but we think it is reasonable. We added this explanation to the manuscript.

278 “The model accurately predicts the number of breeding pairs and number of fledging chicks for both colonies”. This sentence is repetitive and should be removed.

We removed the sentence.

280 the statement “probably due to a lower number of sample counts 281 for Atka Bay than for Pointe Géologie (26 vs 44 counts per season).” Is an assumption based on very little data. With only two sample points, AB and PG, it is impossible to say if this is the case. It would be useful to note this in the text, maybe saying that the lower values at Atka bay could be due to the low sample size, but they could also be due to unknown variance in phenology at the two sites. Further studies at other colonies would confirm these assumptions.

We agree and added the sentence:

“Alternatively the phenology at Atka Bay might be more variable than at Pointe Géologie. Studies at other colonies or further data acquisition at Atka Bay could provide further insights on this issue.”

355 Insert the word optical here: “Optical satellite images, currently used to estimate populations, can only be taken during periods when occupancy is most erratic”

We changed the sentence accordingly.

392-395 I would be a little more cautious with the statement “This increase by 393 more than 2,000 breeding pairs is likely due to an immigration of individuals from the 394 neighboring Halley colony that experienced three consecutive years of breeding failure (from 395 2014 to 2016) and had completely abandoned the colony site in 2016.” I would agree that the displacement of adults from Halley could be the root cause of the increase at Atka, but it is possible that the increase was juvenile recruitment from Halley or displacement of adults from other colonies closer to Halley, where Halley birds had settled. Overall I think it would be more correct to say that the increase in numbers could have been due to immigration, recruitment or displacement, caused by the almost complete abandonment of the Halley Bay colony between 2016 and 2018.

We agree that the immigration of birds directly from Halley to Atka has not been proven, and more complex inter-colony migration is possible. We changed the sentence according to the reviewer’s suggestion:

“This increase by more than 2,000 breeding pairs is likely due to immigration, recruitment or occasional visits of individuals from close by colonies. A possible reason for migration of individuals is the neighboring Halley colony that experienced three consecutive years of breeding failure (from 2014 to 2016) and was completely abandoned in 2016 (23).”

435 On the perspective correction method, do the authors have any statistics on the accuracy of this method compared to drone analysis? It would be useful to get an idea of what the potential error is in the method.

In the cited publication Richter et. al. 2018 (phase transition in huddling emperor penguins) perform a theoretical analysis on the accuracy of the method and find an error of less than 10% for camera-colony distances between 90m and 150m. We do not yet have any comparative drone footage. We agree that such footage would help to validate the method.

465 On the statement: “The most important model parameter is the number breeding males and females. This number is assumed to remain constant during a breeding cycle...” Is there any consideration or analysis of the percentage of non-breeders at the colony? Are all birds breeders? Also, when egg or chick mortality happens surely many breeders leave? At PG there can be over 30% chick mortality (old reports from Prevost), so does this number of breeders really remain constant? I think that this fact is mentioned later in the methods around line 490, but that goes against the original statement, so you may need to change or expand the statement.

The sentence is misleading and has been removed. The model accounts for a higher number of non-breeding birds that leave after a failed courtship period. The number of pairs participating in the breeding also decreases throughout the year due to lost eggs/dead chicks. The model, however, does not account for a low number of non-breeder birds that may be sporadically present at the breeding site (e.g. for molting) and introduce fluctuations to the number of present animals.

Reviewers' Comments:

Reviewer #1:

Remarks to the Author:

The manuscript is much improved, my congratulations to the authors. Their responses are convincing and supported by evidence. The methods are now much clearer. I have a few more suggestions (generally minor) that I hope will improve an already well prepared manuscript.

Line 57: "darkAntarctic" > "dark Antarctic"?

Line 113: ".." > "."

Line 131: The authors do not use the word "estimate" throughout the manuscript much. I don't know if this is intentional. In either case, I think "estimate" makes much more sense than "obtain" when statistical models are involved.

Line 137: Please explicitly specify the seasons (years) here.

Line 205: In the authors response to one of my comments about Fig 4C, they indicate that "we actually don't compute a correlation between the measured and predicted data points displayed", but then they say in the manuscript "The timing of events as estimated by the model shows good agreement with the observed timing of events ($R^2=0.98$, average absolute error of 10 days) when pooled over all years." If you are pooling timing data across multiple breeding events, even if it is also across years, to calculate an R^2 , are you not essentially computing a correlation with what is displayed on fig 4C? I might be missing something here but I think authors need to be more clear about how that super high R^2 was obtained.

Line 314: This statement should either refer to a figure (or a table) on the main text or supplementary information.

Lines 388 and 389: Error in mathematical notation (boxes instead of symbols).

Line 419: I don't find the argument presented here particularly helpful. The authors might be correct in their claim that previous counts could have high uncertainty (which by I think they mean high observation error), but the if their predictions matched with these counts, they would have used it to justify the validity of their model. If the only way for a given set of observations to be correct is for them to match your model, then how can your model ever be wrong? For a more balanced approach please also discuss the possibility that observations are in fact correct and your model can sometimes make erroneous predictions and suggest potential ways forward.

Line 502: I find the use of some of the words here confusing, in terms what is being fit, estimated and derived because a lot of times they are being used interchangeably. I think more standardized approach could make things clearer:

1 - Adult counts from a given year on certain days used as data,

2 - 14 parameters on table 1 are estimated with the phenological model and this data, these parameters include timing related parameters as well as NB and breeding success related parameters,

3 - Numbers of breeding Males, breeding females, non-breeders, and chicks are derived from the estimates of these parameters

While derived parameters are also estimates, I find a more Bayesian jargon here more helpful, where any parameter with a prior is estimated, and the rest is derived. I might be wrong in interpreting the model, in which case please use my suggestions just as a general guideline to standardize some of the statistical jargon.

S9: While this is very helpful for clarifying the model structure, one thing is still missing: the data. Please explicitly indicate how and where manual counts are used in these set of equations when running MCMC sampling.

S23: I'm not sure I understand what is being displayed here. My original suggestion was about

testing the transferability of the phenological model (especially across space). To simplify things here's a suggestion for a test of transferability:

Similar to when you are predicting the colonies with only satellite, fix the 11 parameters of the phenological model to those obtained from PGEO (not Atka Bay), estimate the other three with satellite area/windchill combination from Atka Bay, and make the same predictions you made on Fig 6B for Atka bay manual counts. Repeat this for switching things around between PGEO and Atka Bay (essentially 6A but 11 parameters are fixed to Atka Bay values). I think this way you will have a better understanding of how the transferability of the phenological model is affected when some parameters are fixed to the values of single or a small number of colonies.

Reviewer #2:

Remarks to the Author:

The authors have done a good job of re-organizing and amending this paper as per the suggestions of the two initial reviewers.

I have added a minor number of suggestions and typological corrections in the attached word document.

I would double check for repetition in the reviewed manuscript as adding in some of the suggestions from reviewers may have led to some points being made twice in the new document. One comment that I think needs to be added to the end conclusion paragraph, where the authors suggest expanding the method to all colonies, is that while, in theory, expanding this method out to all emperor colonies may be possible, the cost, both monetarily and in effort, of acquiring 5-10 satellite images of each colony and analyzing them, may be prohibitive with current satellite costing structures and area estimating methodologies.

REVIEWER COMMENTS

Reviewer #1 (Remarks to the Author):

The manuscript is much improved, my congratulations to the authors. Their responses are convincing and supported by evidence. The methods are now much clearer. I have a few more suggestions (generally minor) that I hope will improve an already well prepared manuscript.

Line 57: “darkAntarctic” > “dark Antarctic”?

Thank you for spotting the typo.

Line 113: “..” > “.”

Thank you for spotting the typo.

Line 131: The authors do not use the word “estimate” throughout the manuscript much. I don’t know if this is intentional. In either case, I think “estimate” makes much more sense than “obtain” when statistical models are involved.

We changed “obtain” or similar words for the word “estimate”, where applicable. (lines 131, 140, 145, 215, 252, and 610.)

Line 137: Please explicitly specify the seasons (years) here.

We added the seasons for AB (2018, 2019, 2020) and PG (2014, 2015, 2016, 2017)

Line 205: In the authors response to one of my comments about Fig 4C, they indicate that “we actually don’t compute a correlation between the measured and predicted data points displayed”, but then they say in the manuscript “The timing of events as estimated by the model shows good agreement with the observed timing of events ($R^2=0.98$, average absolute error of 10 days) when pooled over all years.” If you are pooling timing data across multiple breeding events, even if it is also across years, to calculate an R^2 , are you not essentially computing a correlation with what is displayed on fig 4C? I might be missing something here but I think authors need to be more clear about how that super high R^2 was obtained.

Upon re-examining our analysis and your comments, we acknowledge that the R^2 value in relation to the data presented in Fig. 4C was inaccurately expressed in the rebuttal letter (it was correctly expressed in the revised manuscript). It is indeed the coefficient of correlation of the data presented. We thank the reviewer for bringing this point up again. We agree that R^2 score does not significantly contribute to assessing our model's performance in the context presented. Consequently, we have removed the mention of the R^2 value from our discussion of Fig. 4C. In this case the average absolute error is the better and easier to interpret measure of predictive performance.

Line 314: This statement should either refer to a figure (or a table) on the main text or supplementary information.

We added a reference to Fig. 4C and the table of events (S11).

Lines 388 and 389: Error in mathematical notation (boxes instead of symbols).

The symbol should be the Greek letter “rho” which got lost during the pdf generation on the submission platform. We will ascertain that it is properly spelled during the proofing process.

Line 419: I don’t find the argument presented here particularly helpful. The authors might be correct in their claim that previous counts could have high uncertainty (which by I think they mean high observation error), but the if their predictions matched with these counts, they would have used it to justify the validity of their model. If the only way for a given set of observations to be correct is for them to match your model, then how can your model ever be wrong? For a more balanced approach please also discuss the possibility that observations are in fact correct and your model can sometimes make erroneous predictions and suggest potential ways forward.

We agree with the reviewer, that our argument could be worded more accurately. Our aim was not to validate our model or estimates (we used our own ground truth data for this), but to contextualize our estimates and understand the differences to literature values. We reworded the paragraph accordingly:

“Recent satellite based publications (6, 22, 23) as well as this study report rather high discrepancies in the estimated colony population sizes compared to historical aerial and ground based surveys (i.e. Atka Bay -700 animals, Stancomb Wills +3400 animals, Coulman Island -5832 animals). We account these differences to improvements in the estimation of the number of breeding pairs and not to erroneous counts (in either study). Historical ground counts did not account for the phenological status of the colony and simply reported the current number of individuals at the colony, while modern satellite based surveys (and this study) aim to estimate the number of breeding pairs.”

Line 502: I find the use of some of the words here confusing, in terms what is being fit, estimated and derived because a lot of times they are being used interchangeably. I think more standardized approach could make things clearer:

- 1 - Adult counts from a given year on certain days used as data,
- 2 - 14 parameters on table 1 are estimated with the phenological model and this data, these parameters include timing related parameters as well as NB and breeding success related parameters,
- 3 – Numbers of breeding Males, breeding females, non-breeders, and chicks are derived from the estimates of these parameters

While derived parameters are also estimates, I find a more Bayesian jargon here more helpful, where any parameter with a prior is estimated, and the rest is derived. I might be wrong in interpreting the model, in which case please use my suggestions just as a general guideline to standardize some of the statistical jargon.

We thank the reviewer for his proposed clarification in nomenclature.. We changed the verbs fit, estimate, derive, and predict with the appropriate terms throughout the manuscript.

S9: While this is very helpful for clarifying the model structure, one thing is still missing: the data. Please explicitly indicate how and where manual counts are used in these set of equations when running MCMC sampling.

We added a section (“Application to data”), where we explain the process of optimizing the model parameters with the manual count data.

S23: I’m not sure I understand what is being displayed here. My original suggestion was about testing the transferability of the phenological model (especially across space). To simplify things here’s a suggestion for a test of transferability:

Similar to when you are predicting the colonies with only satellite, fix the 11 parameters of the phenological model to those obtained from PGEO (not Atka Bay), estimate the other three with satellite area/windchill combination from Atka Bay, and make the same predictions you made on Fig 6B for Atka bay manual counts. Repeat this for switching things around between PGEO and Atka Bay (essentially 6A but 11 parameters are fixed to Atka Bay values). I think this way you will have a better understanding of how the transferability of the phenological model is affected when some parameters are fixed to the values of single or a small number of colonies.

We replaced S23 in the supplement with a plot like in 6A &B, for models with parameters fixed to the PG and AB values, according to the reviewer suggested procedure. We added the plots for the global parameter (same as in 6 A&B) and a comparative bar plot of the predicted number of breeding pairs and number of fledging chicks, to show the difference between global, AB-, and PG-parameters. The predicted number of breeders and fledging chicks for all years for all three versions lie within a 1-sigma error interval of each other. Hence, we find no significant difference between the model versions.

Reviewer #2 (Remarks to the Author):

The authors have done a good job or re-organizing and amending this paper as per the suggestions of the two initial reviewers.

I have added a minor number of suggestions and typological corrections in the attached word document.

I would double check for repetition in the reviewed manuscript as adding in some of the suggestions from reviewers may have led to some points being made twice in the new document.

We revised our manuscript with regard to the reviewer’s comments and removed unnecessary duplications.

One comment that I think needs to be added to the end conclusion paragraph, where the authors suggest expanding the method to all colonies, is that while, in theory, expanding this method out to all emperor colonies may be possible, the cost, both monetarily and in effort, of acquiring 5-10 satellite images of each colony and analyzing them, may be prohibitive with current satellite costing structures and area estimating methodologies.

We understand that 5-10 images per colony and per year will be costly. However, we consider it worth the effort to know the annual breeding success and not only the number of breeding pairs at a colony. We believe that the availability of satellite images will increase with a number of new constellations being deployed in recent years or about to be deployed, such as World View Legion (launch 2023, 30cm, 15 revisits per day), Pelican (operational 2025, 30 cm, 30 revisits per day), or Albedo (operational 2027, 10cm, 5 revisits per day). We added a paragraph in line 470:

“While it currently might seem cost-prohibitive to expand such an analysis to all colonies, we are convinced that the obtained information and its societal relevance easily outweighs its cost. Upcoming satellite constellations in near future that will be monitoring earth at very high resolution and revisit rate will reduce cost, while an increased revisit rate will improve the possibility to collect high-quality imagery, which, combined with automation and our utilized area detection technique, will drastically simplify analysis, rendering our process feasible.”

Line 352:

This is an interesting point, and leads to the question - when using the phenological model, should satellite abundance estimates be corrected by latitude? My suggestion would be this is an area that needs more ground truthing before we can make this assumption, but it might be worth saying this.

We agree with the reviewer, that estimates from satellite images need to be corrected for the colony latitude. In particular, we already applied this correction in our study of satellite based data by including the latitudinal shift of arrival time. At the end of the discussion of the phenological model, we added:

“This relationship between behavior and latitude helps to increase the precision of our phenological model when animal count data are too sparse to reliably estimate event times such as arrival times. However, more data on the arrival time of breeders at colonies of different latitudes are necessary to consolidate this hypothesis.”

Line 371:

I would also mention that the weather model has a very coarse spatial resolution that does not always model local conditions accurately.

In the case of Atka Bay (2018 - 2020), and Pointe Géologie (2012-2021), we used weather data from the respective research stations Neumayer III, and Dumont d'Urville, as we state in the methods section. The meteorological conditions may still deviate from the sea ice. To acknowledge this, we added to the Manuskript (Lines 372-375):

“Furthermore, local weather conditions can be subject to considerable uncertainty, either because they are extrapolated from data recorded at the nearest weather station (as for the Pointe Géologie and Atka Bay (2018-2020) estimates), or because they are derived from weather models (as for the Stancomb Wills, Coulman Island, and Atka Bay (2011) estimates).”

Line 403:

There are some obvious implications here for long term monitoring. At present we rely on one

satellite count per year at each of the 61 colonies. This is expensive and time consuming. To multiply this by 5-10 times is obviously a burden which you should discuss.

See our comment above on the availability of images.